# FG-PRM: Fine-grained Hallucination Detection and Mitigation in Language Model Mathematical Reasoning

## Abstract

Hallucinations in large language models (LLMs) pose significant challenges in tasks requiring complex multi-step reasoning, such as mathematical problem-solving. Existing approaches primarily detect the presence of hallucinations but lack a nuanced understanding of their types and manifestations. In this paper, we first introduce a comprehensive taxonomy that categorizes the common hallucinations in mathematical reasoning tasks into six types: fabrication, factual inconsistency, context inconsistency, instruction inconsistency, logical inconsistency, and logical error. We then propose FG-PRM (**F**ine-**G**rained **P**rocess **R**eward **M**odel), an augmented model designed to detect and mitigate hallucinations in a fine-grained, step-level manner. To address the limitations of manually labeling training data, we propose an automated method for generating fine-grained hallucination data using LLMs. By injecting hallucinations into reasoning steps of correct solutions, we create a diverse and balanced synthetic dataset for training FG-PRM, which consists of six specialized Process Reward Models (PRMs), each tailored to detect a specific hallucination type. Our FG-PRM demonstrates superior performance across two key tasks: 1) Fine-grained hallucination detection: classifying hallucination types for each reasoning step; and 2) Verification: ranking multiple LLM-generated outputs to select the most accurate solution, mitigating reasoning hallucinations. Our experiments show that FG-PRM outperforms ChatGPT-3.5 and Claude-3 on fine-grained hallucination detection and substantially boosts the performance of LLMs on GSM8K and MATH benchmarks. [1]

## 1 Introduction

While considerable progress has been made in enhancing the general capabilities of large language models (LLMs), solving complex reasoning tasks such as answering mathematical questions remains a challenge. Recently, advanced techniques like Chain-of-Thoughts (Wei et al., 2022), Tree-of-Thoughts (Yao et al., 2024) and Reasoning-via-Planning (Hao et al., 2023) are proposed. These methods guide LLMs in breaking down complex reasoning tasks into simple steps, thus improving their performance and enhancing the interpretability of the reasoning process. However, while generating multi-step reasoning chains can improve performance, LLMs often produce incorrect or unverifiable statements—commonly known as hallucinations—that hinder their ability to solve complex problems that require multiple reasoning steps.

Prior methods of mitigating hallucinations in reasoning chains largely focus on detecting their presence, with limited exploration into the distinct types of hallucinations produced. Our research goes beyond this by developing a fine-grained taxonomy that categorizes hallucinations based on their nature and manifestation (see Figure 1 for an illustration comparing coarse-grained detection with our method). We analyze reasoning steps to pinpoint the emergence of hallucinations and uncover patterns in their behavior.

Recent efforts have shown that training reward models (RMs) is an effective approach for detecting and mitigating hallucinations, with the two primary categories being Outcome Reward Model (ORM) (Cobbe et al., 2021) and Process Reward Model (PRM) (Lightman et al., 2023). ORMs evaluate the correctness

---

*Both authors contributed equally to this work.
[1]Codes and datasets are available at: `https://anonymous.4open.science/r/FG-PRM-75BB/`

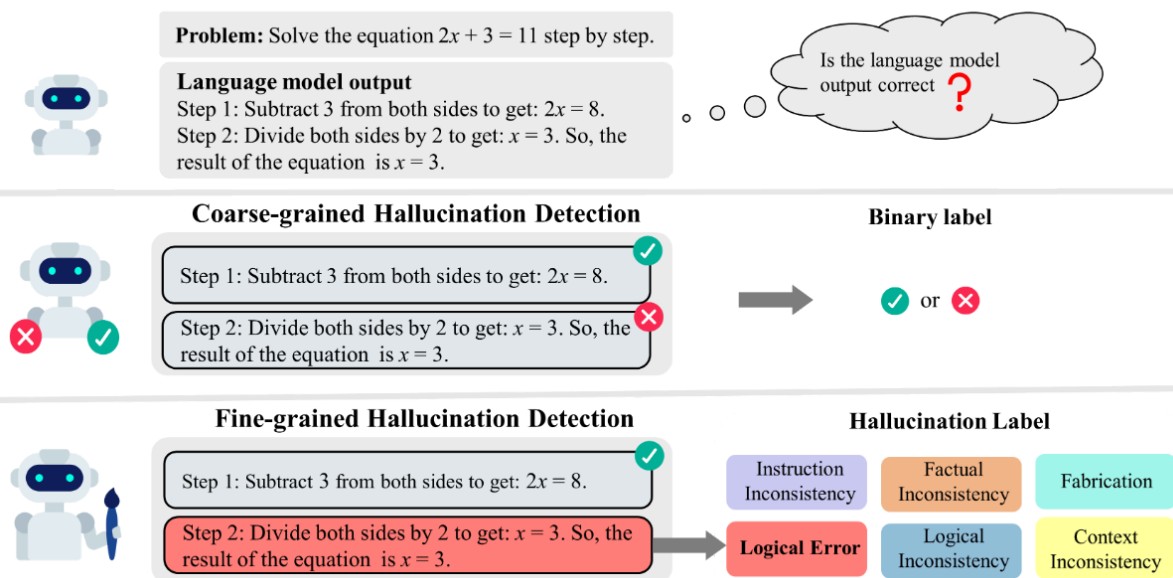

Figure 1: Overview of fine-grained hallucination detection for language model reasoning process. Above is an example for ` Logical Error ` hallucination.

of entire reasoning chains, while PRMs assess each step. PRMs have demonstrated superior performance in many scenarios (Wang et al., 2023) since they can provide more granular feedback and effectively guide models' learning process. However, collecting data to train PRMs is labor-intensive, particularly for complex multi-step reasoning tasks, where human annotation is costly and prone to bias. To address this, we develop a novel method to automatically generate fine-grained hallucination data using LLMs. Specifically, for a given problem with a ground-truth solution, we first identify reasoning steps suitable for hallucination injection. After that, we utilize an LLM to generate additional reasoning steps that incorporate various hallucination types, based on predefined instructions and demonstrations. The generated hallucinatory reasoning steps then serve as negative examples to train task-specific PRMs, each designed to detect a particular hallucination type. This approach improves the accuracy of hallucination mitigation by allowing each PRM to focus on a distinct category.

To evaluate our methods, we test our FG-PRM on two widely used mathematical benchmarks, GSM8K (Cobbe et al., 2021) and MATH (Hendrycks et al., 2021). We validate the effectiveness of our method by two tasks: 1) fine-grained hallucination detection, where we classify different hallucination types at each reasoning step; and 2) verification: where we rank multiple outputs generated by LLMs to select the most accurate solution. Our major contributions are as below:

- We introduce a comprehensive hallucination taxonomy that categorizes common errors in mathematical reasoning tasks into six distinct types.

- We propose an automated method for synthesizing fine-grained hallucination data across all six categories without requiring human annotations. Based on this, we design FG-PRM to detect and mitigate hallucinations in a fine-grained, step-level manner.

- Through extensive experiments, we demonstrate that our FG-PRM surpasses ChatGPT-3.5 and Claude-3 in the hallucination detection task for most hallucination types, achieving over 5% higher F1 scores in these cases. Moreover, FG-PRM trained on our synthetic data, excel on the verification task on both GSM8k and MATH datasets, as compared to PRMs trained on the more costly human-labeled data (Lightman et al., 2023).

## 2   Fine-grained Hallucination Taxonomy

Large language models excel at solving tasks that require complex multi-step reasoning by generating solutions in a step-by-step and chain-of-thought format. Nevertheless, even state-of-the-art models are prone

to inaccuracies, often producing content that is unfaithful, fabricated, inconsistent, or nonsensical. Categorizing and localizing these inaccuracies in reasoning steps is challenging but provides explicit insights into which parts of the model output have specific types of problems. To address the need for a more nuanced understanding of these hallucinations, we propose a refined taxonomy for reasoning tasks.

Building upon the prior work Ji et al. (2023), we develop a fine-grained taxonomy for two major categories of hallucinations: intrinsic and extrinsic hallucination, according to whether the hallucination can be verified by the input information or the contents LLMs have previously generated. To describe more complex errors surfacing in LM reasoning, we further divide the intrinsic hallucination into contextual inconsistency, logical inconsistency and instruction inconsistency, while extrinsic hallucinations are divided into logical error, factual inconsistency, and fabrication. To illustrate our definition of LLM hallucinations more intuitively, we provide examples for each type of hallucination in Appendix Table 4, along with corresponding explanations. The definitions of our proposed categories are elaborated below:

**(1)** **Context Inconsistency** refers to instances where a reasoning step is inconsistent with the contextual information provided by the user.

**(2)** **Logical Inconsistency** refers to the logical contradictions or inconsistencies between the current and previous reasoning steps.

**(3)** **Instruction Inconsistency** refers to instances where a reasoning step does not align with the explicit instructions of the user.

**(4)** **Logical Error** refers to instances where a reasoning step makes incorrect calculations, which should be verified by external information or tools.

**(5)** **Factual Inconsistency** refers to instances where a reasoning step contains facts that can be grounded in real-world information but present contradictions.

**(6)** **Fabrication** refers to instances where a reasoning step contains facts that are unverifiable against real-world knowledge or context information.

Compared to the simplified taxonomy proposed in previous work (Golovneva et al., 2022; Prasad et al., 2023), our refined taxonomy is designed to comprehensively capture the unique complexities of LLM hallucinations, offering a structured framework to study distinct error patterns and enabling more granular analyses and targeted mitigation strategies.

## 3 Task Formulation

In this section, we formulate the two primary tasks of fine-grained hallucination detection and mitigation, highlighting the importance of step-level and fine-grained supervision.

### 3.1 Task 1: Fine-Grained Hallucination Detection

This task aims to detect hallucinations in language model reasoning output at a granular level, focusing on individual reasoning steps. Specifically, the detector is tasked with identifying fine-grained hallucinations in the output of a language model by assigning reward scores for each intermediate step in a reasoning chain. The objective is to classify hallucination types at the step level, determining whether a specific hallucination type is present.

Given a question $x$ and its solution $y$ consisting of $L$ reasoning steps, we assume the ground-truth annotations for hallucination types are available. These annotations, denoted as $y_i^{*t} \in \{\text{TRUE}, \text{FALSE}\}$, provide a binary label for each hallucination type $t$ at the $i$-th step, indicating whether the hallucination $t$ is present (TRUE) or absent (FALSE). The detector models predict $y_i^t$, where $y_i^t$ is the model's predicted label for the $i$-th step and hallucination type $t$. We evaluate the model's performance using standard metrics for classification as in previous work (Feng et al., 2023; Mishra et al., 2024): precision and recall. For each hallucination type $t$, the precision measures the proportion of correct predictions out of all predictions where the model indicated the presence of a hallucination at a step, while recall measures the proportion of actual hallucination steps

that the model correctly identified. These are computed as follows:

$$\text{Precision}^t = \frac{\sum_{i \in L} \mathbb{I}\left[y_i^t = y_i^{*t}\right]}{\sum_{i \in L} \mathbb{I}\left[y_i^t = \text{TRUE}\right]} \tag{1}$$

$$\text{Recall}^t = \frac{\sum_{i \in L} \mathbb{I}\left[y_i^t = y_i^{*t}\right]}{\sum_{i \in L} \mathbb{I}\left[y_i^{*t} = \text{TRUE}\right]} \tag{2}$$

Here, $\mathbb{I}[\cdot]$ is an indicator function that returns 1 if the condition is true and 0 otherwise. Precision indicates the proportion of correctly predicted hallucinations for type $t$, while recall indicates how many of the true hallucinations were detected by the model.

To assess the overall performance across all hallucination types, we calculate the F1 score, which is the harmonic mean of precision and recall. The F1 score is computed for each hallucination type and then averaged across all types $\mathcal{E}$:

$$\text{F1 Score} = \frac{1}{|\mathcal{E}|} \sum_{t \in \mathcal{E}} \frac{2 \times \text{Precision}^t \times \text{Recall}^t}{\text{Precision}^t + \text{Recall}^t} \tag{3}$$

Thus, fine-grained hallucination detection can be framed as a set of binary classification tasks, where the system predicts whether each reasoning step $s_i$ contains a specific hallucination type. By evaluating precision, recall, and F1 score across different hallucination types, we gain a comprehensive understanding of the model's ability to detect and categorize hallucinations within complex reasoning processes.

### 3.2 Task 2: Fine-Grained Hallucination Mitigation

The verification task (Lightman et al., 2023) assesses a model's ability to evaluate and rank multiple candidate solutions for a given problem. In this task, a generator produces $N$ possible solutions $\{y^1, y^2, \ldots, y^N\}$ for a problem $x$, which are then evaluated by a reward model (Section 4.1). The reward model assigns a score to each candidate solution based on its correctness, with the goal of selecting the best solution among the candidates.

This task follows the best-of-N selection method, where the solution with the highest score is chosen as the final answer. A well-performing reward model improves the likelihood of selecting the correct solution, thereby enhancing the overall problem-solving accuracy. By providing meaningful feedback on each candidate solution, the verification task helps ensure that the reasoning process is grounded in correctness and consistency.

## 4 Methodology

In this section, we first introduce two basic types of reward models (Section 4.1), the Outcome Reward Model (ORM) and the Process Reward Model (PRM). After that, we describe our automated framework for generating hallucination-annotated datasets, followed by a detailed explanation of the training procedure for our Fine-Grained Process Reward Model (FG-PRM), elaborating on the use of generated datasets and how our model enhances both hallucination detection and verification performance (Section 4.2).

### 4.1 Preliminary

**ORM** The Outcome-supervised Reward Model (ORM) was introduced by Cobbe et al. (2021). Given a problem $x$ and its solution $y$, an ORM assigns a sigmoid score $r_y$ to the entire solution, indicating whether $y$ is correct. ORMs are typically trained using cross-entropy loss over the entire solution. Assume $y^*$ is the ground-truth label of the solution $y$, $y^* = 1$ if $y$ is correct, otherwise $y^* = 0$. The training objective minimizes the cross-entropy between the predicted outcome $r_y$ and the ground-truth $y^*$:

$$\mathcal{L}_{\text{ORM}} = y^* \log r_y + (1 - y^*) \log(1 - r_y) \tag{4}$$

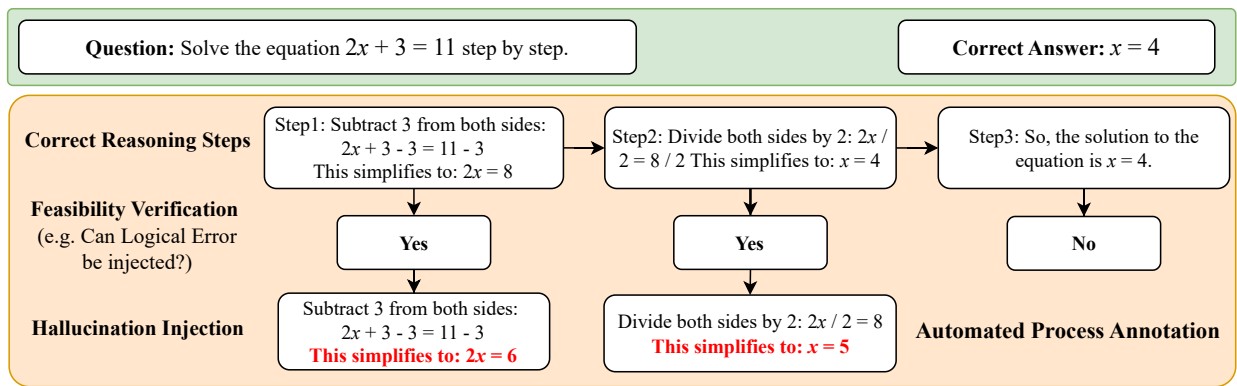

Figure 2: Our automated reasoning process annotation involves two steps: First, for each reasoning step, we instruct a language model to verify the feasibility of injecting hallucinations (using Logical Error as an example in this figure). Second, for steps where hallucinations can be injected, we prompt the language model to introduce hallucinations by providing instructions (see in Appendix F) and few-shot demonstrations (see in Appendix G), which serve as negative examples for training FG-PRM .

However, ORM's coarse feedback mechanism limits its ability to diagnose errors within individual reasoning steps, as it only evaluates the final solution without considering intermediate correctness.

**PRM** The Process-supervised Reward Model (PRM), introduced by Lightman et al. (2023), addresses the limitations of ORM by providing fine-grained, step-level feedback. Instead of assigning a single score to the entire solution, PRM assigns a sigmoid score $r_{y_i}$ for each reasoning step $y_i$ in the solution $y$. This approach enables the model to evaluate the correctness of each intermediate step, providing more detailed feedback on where the reasoning process succeeds or fails. The training objective for PRM minimizes the sum of cross-entropy losses over all reasoning steps, allowing the model to learn from fine-grained supervision:

$$\mathcal{L}_{\mathrm{PRM}} = \sum_{i=1}^{L} \log y_i^* \log r_{y_i} + (1 - y_i^*) \log(1 - r_{y_i}) \tag{5}$$

where $L$ is the number of reasoning steps in the solution $y$ and $y_i^*$ is the ground-truth label of the $i$-th step of $y$. By providing feedback at the step level, PRM offers significant advantages over ORM in tasks requiring complex, multi-step reasoning. PRM not only improves the model's ability to detect and correct errors within individual steps but also enables more targeted learning and fine-tuning.

### 4.2 FG-PRM: Fine-grained Process Reward Model

In this Section, we introduce our FG-PRM, the **F**ine-**G**rained **P**rocess **R**eward **M**odel for hallucination detection and mitigation. To reduce the annotation cost issues associated with PRM, we first introduce an automated process annotation framework for step-level fine-grained dataset synthesis. After that, we provide the training details for our FG-PRM on the synthetic dataset.

#### 4.2.1 Automated Hallucination Generation

To detect fine-grained hallucinations in language model reasoning tasks, we propose a framework based on fine-grained, step-level process supervision. Existing step-level datasets with fine-grained annotations (Golovneva et al., 2022) are limited in size, and collecting the necessary data for training models with such detailed labels is costly, as it requires human annotators to provide fine-grained feedback for each reasoning step. To overcome the scarcity of human-labeled data, we introduce an automated reasoning process hallucination annotation framework, as illustrated in Figure 2. We treat the golden chain-of-thought (CoT) reasoning steps as positive examples, while negative examples are generated by injecting hallucinations into these steps using Llama3-70B (Dubey et al., 2024). To synthesize the negative examples, we adopt a two-step process as follows.

**Step 1: Identify where to inject hallucination** In our taxonomy, each hallucination type has distinct characteristics, requiring specific conditions and methods for generation. However, not all reasoning steps are

suitable for generating every type of hallucination. For instance, when a reasoning step is exclusively focused on numerical calculations, it becomes challenging to introduce factual inconsistency. To effectively introduce a hallucination type into the reasoning process, we need to identify steps that meet the necessary conditions for hallucination generation. To achieve this, we have developed a set of tailored rules within the prompts for the Llama3-70B model. These rules guide the model in determining whether a given reasoning step provides the elements required for a specific type of hallucination. For example, when evaluating whether a step can introduce factual inconsistency, the model checks if the reasoning step references objects (e.g., quantities, features) or named entities. This enables us to manipulate the information, allowing for the seamless integration of contextual inconsistencies in later steps. The complete set of rules for identifying hallucination injection points across the six hallucination types is detailed in Appendix E.

**Step 2: Hallucinate the ground truth reasoning steps**   To control the distribution of hallucinations in the generated dataset and improve the success rate of incorporating our hallucination taxonomy, we prompt the Llama3-70B model to insert hallucinations one by one from our taxonomy. We begin by inputting specific instructions for each hallucination type into the system prompt, guiding the model on how to modify the reasoning process and introduce the desired hallucination. Detailed instructions for each hallucination type are provided in Appendix F. Next, we employ an in-context learning strategy by providing two demonstrations for each query type. Each demonstration includes an example of an injected hallucination, along with an explanation of how it is introduced. These demonstrations can be found in Appendix G. After confirming the appropriate location for injecting the hallucination, we present the problem and the correct reasoning history to the model, instructing it to generate the next reasoning step with the target hallucination. For cost efficiency, we delegate the task of hallucinating reasoning steps to the Llama3-70B model. We experimentally found that our method enables the language model to consistently generate hallucinatory reasoning steps with a high success rate.

### 4.2.2   Model Training

After generating six types of hallucination datasets using our automated data annotation method, we train our FG-PRM, denoted as $R_\Phi$, which comprises six distinct PRMs, $R_{\phi_1} \ldots R_{\phi_6}$, each corresponding to a specific type of hallucination in our taxonomy.

Formally, given an input question $x$ and the corresponding solution $y$ composed of $L$ reasoning steps $\{y_1, y_2, \ldots, y_L\}$, we separately train task-specific PRMs $R_{\phi_t}$ to detect whether each reasoning step in $y$ contains the hallucination type $t$.   The model input has the format of "`question:` $q$, `reasoning steps:` $y_1$ `[sep]` $y_2$ `[sep]` $\ldots y_L$ `[sep]`", where each `[sep]` token represents the classification output at each reasoning step to indicate whether the previous step $y_i$ contains the hallucination type $t$. We define $R_{\phi_t}(x, y_i) = P(\texttt{[sep]} = +1)$ to represent the probability that the step $y_i$ contains the hallucination type $t$. To train each PRM $R_{\phi_t}$, we utilize a step-level classification loss as in Eq.5 to each `[sep]` token before step $y_i$. Overall, our FG-PRM $R_\Phi$ generates an aggregate reward for the solution $y$ of the input question $x$:

$$R_\Phi(x, y) = \sum_{t=1}^{6} \sum_{i=1}^{L} \log\left(R_{\phi_t}(x, y_i)\right) \tag{6}$$

In the verification task, the log-sum of these probabilities is used to aggregate the rewards, resulting in the final reward assigned by FG-PRM for a solution. Importantly, if a step is correct, its probability of correctness is close to 1, contributing minimally to the aggregated reward. This ensures that the length of the answer does not influence the final reward score, maintaining fairness regardless of the sequence length.

Table 1: Performance of fine-grained hallucination detection across all hallucination types on synthetic data and human-annotated data. All numbers are F1 scores.

| Detector | Synthetic Reasoning Chain | | | | | | | Human-annotated Reasoning Chain | | | | | | |
| --- | --- | --- | --- | --- | --- | --- | --- | --- | --- | --- | --- | --- | --- | --- |
| | CI | LI | II | LE | FI | FA | Avg. | CI | LI | II | LE | FI | FA | Avg. |
| ChatGPT | 0.415 | 0.522 | 0.453 | 0.360 | **0.428** | 0.900 | **0.513** | 0.442 | 0.552 | 0.510 | **0.377** | **0.487** | **0.840** | **0.531** |
| Claude | 0.448 | 0.388 | 0.493 | 0.275 | 0.373 | **0.963** | 0.490 | 0.434 | 0.460 | 0.478 | 0.359 | 0.428 | 0.758 | 0.503 |
| PRM | 0.399 | 0.455 | 0.467 | **0.402** | 0.358 | 0.565 | 0.441 | 0.394 | 0.493 | 0.484 | 0.357 | 0.403 | 0.435 | 0.428 |
| FG-PRM | **0.488** | **0.549** | **0.529** | 0.398 | 0.422 | 0.608 | 0.499 | **0.526** | **0.575** | **0.513** | **0.377** | 0.426 | 0.484 | 0.484 |

## 5 Experiments

### 5.1 Settings

**Datasets** We conduct our experiments on two widely used mathematical benchmarks, GSM8K (Cobbe et al., 2021) and MATH (Hendrycks et al., 2021). GSM8K is a high-quality dataset consisting of grade school math problems designed to benchmark the reasoning abilities of language models. To construct the hallucinatory reasoning steps, we employ a meta-dataset and software library (Ott et al., 2023), which collects the golden chain-of-thought solutions for each problem in the GSM8K. MATH, on the other hand, is a large-scale dataset designed for probing and improving model reasoning, which includes human-written step-by-step solutions.

Following Lightman et al. (2023), Uesato et al. (2022) and Wang et al. (2023), we sample instances from GSM8K and MATH datasets to build training sets and test sets. For GSM8K, we randomly sample 700 instances from the training set and 100 instances from the test set. For MATH, we randomly sample 700 instances from the training set. The test set is sampled from the MATH dataset and includes 100 instances. We call both datasets "Coarse-grained Hallucinations" (CG-H). They include human-annotated reasoning chains. Each step has a binary label indicating its correctness. Based on the two sampled datasets, we augment each to 12,000 instances by our automatic hallucination generation method described in Section 4.2.1, including all types of hallucinations described in Section 2 with a balanced hallucination distribution. The augmented datasets are called "Fine-grained Hallucinations" (FG-H). Additionally, we sample 12,000 instances from Math-Shepherd (Wang et al., 2023). This dataset includes automatically constructed process-wise supervision data using the Monte Carlo tree search method, where each step is assigned a binary label to indicate its correctness. For the hallucination detection task, we construct both synthetic and human-annotated datasets based on questions from the MATH dataset. The CG-H (MATH) and FG-H (MATH) datasets are used to train PRM and FG-PRM models, respectively, and to evaluate their performance. For the hallucination mitigation task, we employ the complete CG-H and FG-H datasets, along with Math-Shepherd, to train various reward models. Across all datasets, the training-to-testing split ratio is maintained at 95:5.

**Models** In the fine-grained hallucination detection task, we evaluate the performance of prompt-based and model-based detection. For prompt-based detection, we apply ChatGPT (GPT-3.5-turbo-0125) (Ouyang et al., 2022) and Claude (Claude-3-haiku)[2] with carefully designed prompts as baseline methods. For the model-based detection, we compare our FG-PRM with the traditional coarse-grained PRM (Lightman et al., 2023).

In the fine-grained hallucination mitigation task, we apply various verifiers to evaluate the correctness of solutions generated by language models (generators). We employ Llama3-70B (Dubey et al., 2024) as our solution generator, from which we sample 64 candidate solutions for each test problem. We apply the LongFormer-base-4096 (Beltagy et al., 2020) and Llama-3-8B (Dubey et al., 2024) as our base models due to its strong performance in handling long-context reasoning. We do not modify their main structure in our experiments. In detail, to build a reward model, we replace the output layer with an MLP layer to predict binary hallucination labels. Verifiers include self-consistency (SC), ORM, PRM, and FG-PRM. The self-consistency verifier serves as a baseline without specific model training; it aggregates multiple reasoning paths and selects the most frequent solution as the final answer. Both ORMs and PRMs are trained on the CG-H dataset. For our FG-RPM, we train individual fine-grained PRMs for each of six hallucination types, following the same supervision as PRMs on FG-H data. We conduct all of our experiments using four Nvidia A100 80G GPUs.

---

[2]https://claude.ai/

Table 2: Performance of different verifiers on GSM8K and MATH benchmarks. The evaluation is based on 64 candidate solutions generated by Llama3-70B model with greedy decoding for each problem. Each result is the mean of results from 3 groups of sampling results. Statistical significant test on most improvements compared to the "Self-Consistency" have ($p < 0.05$). Data with $^\dagger$ mark indicate the significant test with $p < 0.05$ is not passed.

| Base Model | Verifier / Reward Model | GSM8K | MATH |
|---|---|---|---|
| - | Self-Consistency | 0.88 | 0.48 |
| LongFormer | ORM | 0.88$^\dagger$ | 0.51 |
| | PRM | 0.89 | 0.53 |
| | Math-Shepherd (ORM) | 0.90 | 0.52 |
| | Math-Shepherd (PRM) | 0.91 | 0.54 |
| | FG-PRM (Ours) | **0.94** | **0.57** |
| Llama3-8B | ORM | 0.87$^\dagger$ | 0.52 |
| | PRM | 0.90 | 0.53 |
| | Math-Shepherd (ORM) | 0.89 | 0.51 |
| | Math-Shepherd (PRM) | 0.91 | 0.53 |
| | FG-PRM (Ours) | **0.93** | **0.58** |

## 5.2 Hallucination Detection Results

To evaluate the efficacy of our method in detecting fine-grained hallucinations, we conduct two experiments on synthetic and human-annotated data.

**Synthetic Data**  We utilize the automated annotation labels from our synthetic dataset, FG-H (MATH), as the golden standard for evaluating various detectors across six types of hallucinations.

As shown in Table 1, FG-PRM outperforms prompt-based detectors in detecting CI , LI , II , and LE , demonstrating that FG-PRM has effectively learned the patterns of these hallucinations and is capable of detecting them accurately. Notably, FG-PRM outperforms PRM in detecting all types of hallucinations, demonstrating the advantages of the fine-grained detection manner. However, prompt-based detectors outperform FG-PRM on FI and FA , primarily due to their larger model sizes and greater access to fact-based knowledge. This reflects the inherent advantage of large language models in fact-based verification. Moreover, precision and recall results are in Tables 5 and 6 in Appendix B. Besides the six individual verifiers in FG-PRM, we also conduct experiments on a single multi-class verifier, which performs worse than our FG-PRM. Additional results are shown in Appendix C.

**Human-annotated Data**  To further validate the effectiveness of our method on real-world data, we conduct an evaluation using human-annotated data. Specifically, for each hallucination type, we first utilize ChatGPT (GPT-3.5-turbo-0125) (Ouyang et al., 2022) to generate step-by-step solutions for 50 problems from the MATH dataset. These solutions are then manually annotated by three graduate students according to the taxonomy of hallucination types proposed in Section 2, ensuring that each selected solution included at least one step exhibiting the target hallucination type. This process resulted in a human-labeled dataset covering six hallucination types, with each type represented by 50 annotated responses corresponding to 50 problems, featuring step-level hallucination labels. The annotations achieved a mutual agreement rate of 79% among the annotators.

The results on the human-annotated data align closely with the trends observed on the synthetic data. Our FG-PRM model demonstrates improving performance in detecting CI and LI hallucinations, where it consistently outperforms both ChatGPT and Claude. However, FG-PRM's performance is slightly below that of the strong, non-public LLMs (e.g. ChatGPT and Claude) in detecting FI and FA hallucinations. This is largely attributable to FG-PRM's smaller parameter size and limited access to world knowledge. Despite these challenges, FG-PRM performs competitively overall, particularly in reasoning-related hallucinations. Further analysis about reasoning chain evaluation for various verifiers is presented in Appendix D.

### 5.3 Hallucination Mitigation Results

Table 2 presents a performance comparison of various verifiers on GSM8K and MATH. FG-PRMs trained on our augmented dataset, FG-H, significantly outperform all baselines across both base models. Notably, after fine-tuning with FG-H, Longformer and Llama3-8B achieve 94% and 58% accuracy on GSM8K and MATH, respectively, surpassing PRMs trained on Math-Shepherd. The results show that base models mitigated by PRMs consistently outperform those mitigated by ORMs, consistent with findings from Uesato et al. (2022), Lightman et al. (2023), and Wang et al. (2023). On GSM8K, most baseline verifiers perform close to the self-consistency level due to the simplicity of the dataset, where many questions involve only basic arithmetic operations. However, the differences between verifiers become more evident in the more complex MATH dataset, where questions and reasoning steps often require LaTeX math expressions. These results indicate that the balanced fine-grained step-level supervision employed by FG-PRMs offers a more robust and effective approach to hallucination mitigation, particularly in handling complex problem-solving tasks.

## 6 Analysis

**Hallucination Mitigation Performance with Different Number of Candidate Solutions**  Figure 3 illustrates the performance of four verifiers with the number of candidate solutions ranging from 1 to 64 across two benchmarks. This demonstrates that FG-PRM consistently outperforms all other verifiers. With predicted insights, the performance gap between FG-PRM and other baseline verifiers will increase with the growth of N.

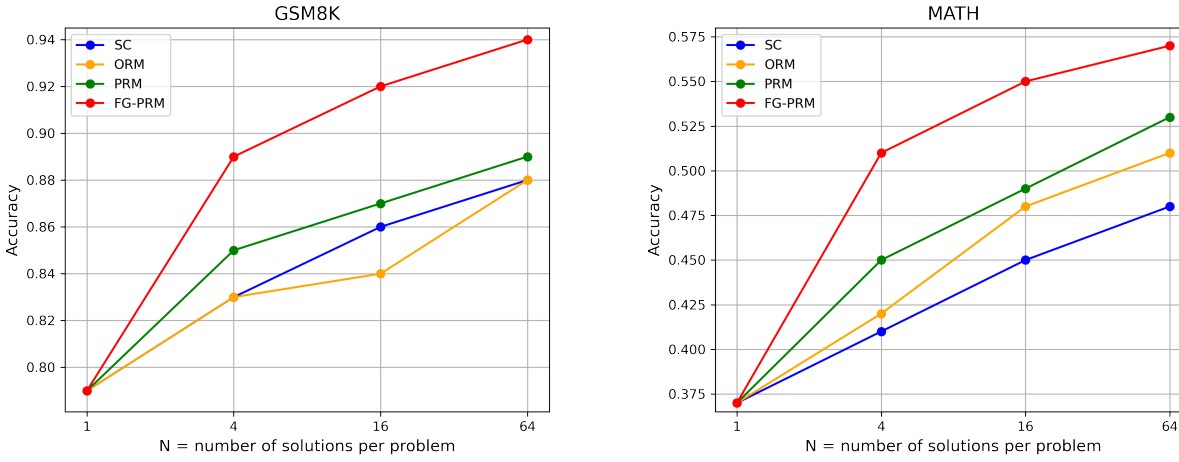

Figure 3: Performance of LLaMA3-70B using different verification methods across different numbers of candidate solutions on GSM8K and MATH.

**Out-of-Distribution Dataset Evaluation**  We further conduct out-of-distribution (OOD) evaluation experiments to assess the robustness and transferability of our approach. In these experiments, we train the PRM verifier on CG-H (GSM8K) and FG-PRM verifier on FG-H (GSM8K). For comparison, we also train them on CG-H (MATH) and FG-H (MATH). Finally, we test all verifiers on the MATH dataset. Notably, the GSM8K dataset contains simple questions, predominantly solvable through basic arithmetic operations, in contrast to the more complex MATH dataset.

As detailed in Figure 4, the performance gap for FG-PRM (+0.01) is slighter than PRM (+0.03). Moreover, the verifiers trained on the FG-H (GSM8K) demonstrate performance closely comparable to those trained on the FG-H (MATH) dataset. This indicates that the FG-PRM verifier trained on FG-H effectively learns to recognize patterns of hallucinations

Figure 4: Out-of-distribution performance on the MATH benchmark.

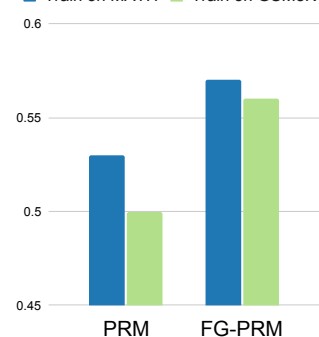

and can generalize this knowledge to tackle more challenging scenarios effectively.

**Qualitative Result of Fine-Grained Hallucination Evaluation**   To illustrate the effect of our FG-PRMs, we demonstrate a case study in Table 3, comparing the hallucination detection results of FG-PRM and CG-PRM. Our FG-PRM demonstrates remarkable discrimination by precisely detect fine-grained types of hallucination in reasoning steps. Notably, Step 7 has a high ⬛Logical Inconsistency score since the step incorrectly refers to the formula in previous steps. It refers to $a = 5$ as $a = 11$, which causes ⬛Logical Inconsistency. FG-PRM recognizes it by giving a low score, while CG-PRM does not recognize this detail.

Table 3:   A case study from the MATH dataset. A high CG-PRM score indicates that the step is positive. A low FG-PRM-TYPE score indicates the step has a high probability of having the TYPE of hallucination.

| | |
|---|---|
| **Question** | *Each edge length of a rectangular solid is a prime number. If the volume of the rectangular solid is 385 cubic units, what is the total surface area, in square units, of the rectangular solid?* |
| **Answer** | 334 |
| **Reasoning Steps** | Step 1:We know that the volume of this rectangular solid is the product of its three side lengths. FG-PRM: 0.95, CG-PRM: 0.83. 
 Step 2: Let's call those side lengths $a$, $b$, and $c$. The volume is $abc$. FG-PRM: 0.83, CG-PRM: 0.80. 
 ...... 
 Step 6: Let's plug in the possible values. If $a = 5$, $b = 7$, and $c = 11$, then $2ab + 2bc + 2ac = 2(5)(7) + 2(7)(11) + 2(5)(11) = 74 + 154 + 110 = 338$. FG-PRM: 0.87, CG-PRM: 0.72. 
 Step 7: Which simplifies to $14 \cdot 11 + 22 \cdot 11 + 10 \cdot 11$. FG-PRM: Logical Inconsistency 0.07 CG-PRM: 0.80. 
 Step 8: So the total surface area would be 334. FG-PRM : 0.88 CG-PRM: 0.66. |

## 7   Related Work

**Hallucinations in LLM**   Numerous studies have explored the taxonomy of hallucinations in language models, offering valuable insights. Mishra et al. (2024) identify six fine-grained types of factual hallucinations in information search scenarios. Zhang et al. (2023) classify hallucinations based on conflict types: input-conflicting, context-conflicting, and fact-conflicting. Huang et al. (2023a) categorize hallucinations into factuality and faithfulness types, subdividing them further. Ji et al. (2023) discussed intrinsic and extrinsic hallucinations, focusing on whether outputs contradict input content. Our taxonomy, in contrast, concentrates on hallucinations in reasoning tasks, particularly through chain-of-thought reasoning. Closest to our work, Golovneva et al. (2022) propose ROSCOE, measuring semantic alignment, similarity, and language coherence in reasoning chains. Unlike ROSCOE that includes common issues like grammar, redundancy, and repetition, which are easier to spot in general, our taxonomy provides detailed distinctions between error types, providing a compact framework for diagnosing complex errors and improving model outputs.

**Evaluation of Reasoning Chains**   Depending on whether requiring golden references, methods to evaluate reasoning chains can be roughly divide as reference-dependent and reference-free ones. For reference-dependent, the reasoning chains can be evaluated with LLMs (Ren et al., 2023; Adlakha et al., 2023)), or by measuring the discrepancy between the vanilla response and reference (Huo et al., 2023; Pezeshkpour, 2023). For reference-free metrics, some methods rely on aggregating the individual token probabilities assigned by the LLM during generation so that they can reflect reasoning chain uncertainty (Manakul et al., 2023; Huang et al., 2023b). In addition to that, many model-based methods have emerged to evaluate reasoning chains. He et al. (2024) proposed to prompt GPT-4 in a Socratic approach. Hao et al. (2024) employ GPT-4 to summarizes evaluation criteria tailored for each task, and then evaluate the reasoning chains following the criteria. In this work, we focus on model-based reference-free reasoning chain evaluation from the perspective of hallucination detection.

**Improving reasoning abilities of LLMs** For LLMs that have completed training, prompting techniques is an effective approach to improve the performance of LLMs on reasoning tasks without modifying the model parameters. Many studies have developed different prompting strategies in reasoning tasks, such as the Chain-of-Thought Wei et al. (2022); Fu et al. (2022), Tree-of-Thoughts Yao et al. (2024). Besides, instead of directly improving the reasoning performance of LLMs, verifiers can raise the success rate in solving reasoning tasks by selecting the best answer from multiple decoded candidates. Two types of verifiers are commonly used: Outcome Reward Model (ORM) and Process Reward Model (PRM). PRM provides a more detailed evaluation by scoring each individual step. However, training a PRM requires access to expensive human-annotated datasets, which can be a barrier to the advancement and practical application of PRM. To overcome this challenge, methods such as Math-Shepherd Wang et al. (2023) and MiPS Wang et al. (2024) have explored Monte Carlo estimation to automate the data collection process without human involvement, and OmegaPRM Luo et al. (2024) proposed a divide-and-conquer style Monte Carlo tree search algorithm for automated process supervision data generation. Different from the above methods, we directly generate negative examples with hallucinations instead of annotating the reasoning process.

## 8 Conclusion

In conclusion, we propose FG-PRM framework, which introduces a nuanced approach for comprehensive understanding and mitigations of hallucinations in language model reasoning, which are categorized into six distinct types under our new paradigm. By leveraging a novel automatic data generation method, we significantly reduce the dependency on costly human annotations while enriching the dataset with diverse hallucinatory instances. Our empirical results demonstrate that FG-PRM, when trained on this synthetic data, significantly enhances the accuracy of hallucination detection, providing an effective mechanism for improving the LLM reasoning accuracy and faithfulness.

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

## Appendix

## A    Examples of Fine-Grained Hallucination

Table 4: Examples of various types of LLM hallucinations in our taxonomy. Content marked in Red represents the hallucinatory output, while content marked in Blue indicates user instruction or provided context that contradicts the LLM hallucination.

| Hallucination Type | Sub-Type | User Input | Model Output | Explanation |
|---|---|---|---|---|
| Extrinsic Hallucinations | Factual Inconsistency | What is the capital of Australia? | The capital of Australia is Sydney. | The model incorrectly states that Sydney is the capital of Australia, while the correct capital of Australia is Canberra. This error demonstrates factual inconsistency as the model's response contradicts the real-world fact. |
| | Fabrication | What is the population of Atlantis? | The population of Atlantis is approximately 1.2 million people. | The model provided a specific population figure for Atlantis, a fictional place with no basis in reality. Since Atlantis does not exist, there is no verifiable population data. |
| | Logical Error | If a rectangle has a length of 4 units and a width of 3 units, what is the perimeter of the rectangle? | The perimeter formula of a rectangle is: Perimeter = 2 × (length + width). The length is 4 units and the width is 3 units. So, the perimeter would be 2 × (4 + 3) = 16. So the answer is 16 units. | In the red reasoning step, the model incorrectly calculates 2 × (4 + 3) = 16, which should be 14. The error should be verified by external tools like calculators, which matches the definition of Logical Error hallucination. |
| Intrinsic Hallucination | Instruction Inconsistency | Please calculate the quotient of 25 divided by 100 and represent the result as a fraction. | The quotient of 25 divided by 100 is 0.25. | Although the model correctly calculates the result 0.25, it fails to follow the "as a fraction" instruction in the user input. |
| | Context Inconsistency | Given that I have 5 apples and I give 2 to my friend, how many apples do I have left? | Considering the information provided, if you initially had 5 apples and you gave 3 to your friend, you would have 2 apples left. | In the red reasoning step, the model incorrectly refers to the number of apples in the context. The user stated that they gave 2 apples to their friend, but the model incorrectly refers to it as 3. |
| | Logical Inconsistency | How many hours are there in a week if each day has 24 hours? | There are 7 days in a week. Since each day has 24 hours, a week would have 6 days times 24 hours, which equals 144 hours. | The model correctly states that there are 7 days in a week but then incorrectly calculates the number of hours by multiplying 6 days by 24 hours to get 144 hours. This shows a contradiction between the days in a week referred and a failure to maintain internal logical consistency in the reasoning process. |

## B  Detailed Fine-grained Hallucination Detection Results

The precision and recall of the fine-grained detection results for the Llama3-70B generation are reported in Table 5 and 6, respectively.

Table 5: Precision for fine-grained hallucination detection across different categories.

| Detector | Hallucination Type | | | | | | Average |
|---|---|---|---|---|---|---|---|
| | CI | LI | II | LE | FI | FA | |
| ChatGPT | 0.403 | 0.488 | 0.450 | 0.424 | **0.412** | 0.890 | **0.511** |
| Claude-3 | 0.417 | 0.368 | 0.490 | 0.248 | 0.357 | **0.952** | 0.472 |
| PRM | 0.393 | 0.421 | 0.443 | 0.324 | 0.374 | 0.527 | 0.414 |
| FG-PRM | **0.428** | **0.513** | **0.528** | **0.413** | 0.403 | 0.589 | 0.479 |

Table 6: Recall for fine-grained hallucination detection across different categories.

| Detector | Hallucination Type | | | | | | Average |
|---|---|---|---|---|---|---|---|
| | CI | LI | II | LE | FI | FA | |
| ChatGPT | 0.440 | 0.600 | 0.460 | 0.541 | **0.477** | 0.920 | **0.573** |
| Claude-3 | 0.525 | 0.433 | 0.500 | 0.334 | 0.416 | **0.990** | 0.533 |
| PRM | 0.415 | 0.498 | 0.493 | 0.541 | 0.352 | 0.615 | 0.486 |
| FG-PRM | **0.571** | **0.597** | **0.560** | **0.546** | 0.462 | 0.635 | 0.562 |

## C  Compact FG-PRM Verifier

Besides six individual binary classifier verifiers in our FG-PRM, we train a single multi-class verifier on a Longformer model, denoted as FG-PRM (compact). For this reward model, we replace the output layer with an MLP layer that produces seven category outputs, covering six types of hallucinations and a "no error" category. As shown in Tables 7 and 8, the performance of our separate FG-PRMs surpasses that of the compact experimental setting in both the hallucination detection and mitigation tasks.

Our findings indicate a tendency for the model to predict the "no error" label. This bias is primarily due to the imbalance in the training data. Instances with a specific type of hallucination account for only one-sixth of the entire dataset. Additionally, in the multi-step reasoning process, only a few steps display specific hallucinations. As a result, models can achieve high accuracy by predominantly predicting "no error."

Table 7: F1 for fine-grained hallucination detection across different categories.

| Detector | Hallucination Type | | | | | | Average |
|---|---|---|---|---|---|---|---|
| | CI | LI | II | LE | FI | FA | |
| FG-PRM (Compact) | 0.402 | 0.493 | 0.481 | 0.378 | 0.371 | 0.574 | 0.450 |
| FG-PRM | 0.488 | 0.549 | 0.529 | 0.398 | 0.422 | 0.608 | 0.499 |

## D  Reasoning Step Hallucination Evaluation

We utilize our model to evaluate hallucination issues in the generated outputs of large language models. Each generation is assigned six scores corresponding to hallucination types. The score under each hallucination type for a model is calculated based on the proportion of correct reasoning steps in generations. Specifically, score $= \frac{1}{N} \sum_{i=1}^{N} \frac{\# \text{ of correct step}}{\# \text{ of total step}}$ , where $N$ is the total number of generations in the test set. A model with high scores indicates fewer hallucination issues in its generation.

Table 8: Performance of FG-PRM and FG-PRM Compact verifiers on GSM8K and MATH benchmarks. The evaluation is based on 64 candidate solutions generated by Llama3-70B model with greedy decoding for each problem. Each result is the mean of results from 3 groups of sampling results.

| Verifier / Reward Model | GSM8K | MATH |
|---|---|---|
| Self-Consistency | 0.88 | 0.48 |
| FG-PRM (Compact) | 0.90 | 0.54 |
| FG-PRM (Ours) | **0.93** | **0.58** |

Similar to the hallucination mitigation task, we apply our verifiers on Llama3-70B to help it select the best generation among 64 options. The performance is shown in Figure 5. Llama3-70B, with help from verifiers, performs better than itself. The performance trend under each hallucination type aligns well with the results in Table 2 that FG-PRM performs the best among all verifiers.

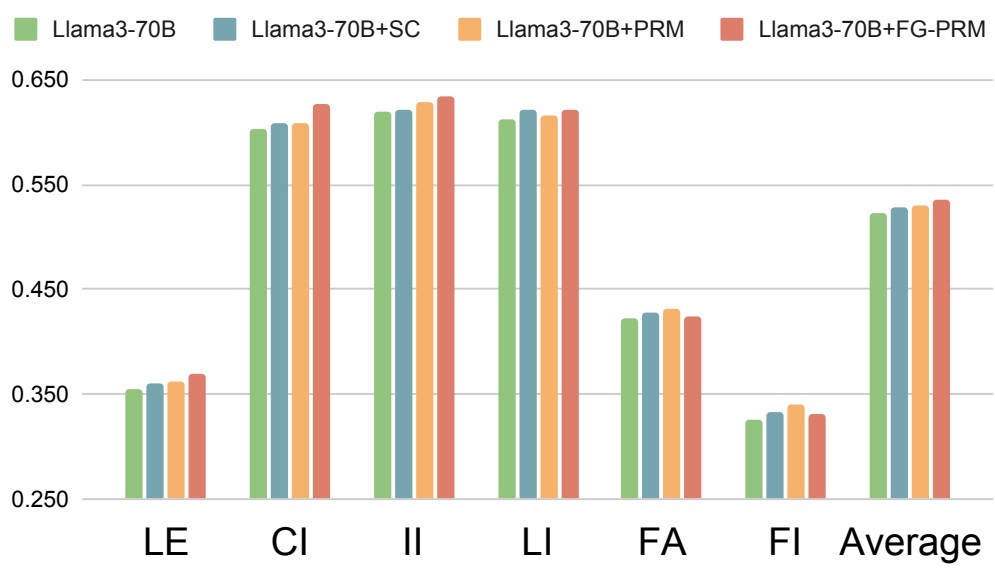

Figure 5: Hallucination Evaluation Performance on Various Models with Verifiers.

## E  Tailored Rules for Judging Hallucination Types

We provide a prompt template for a language model to judge if the reasoning history of a given question can be incorporated into a specific type of hallucination:

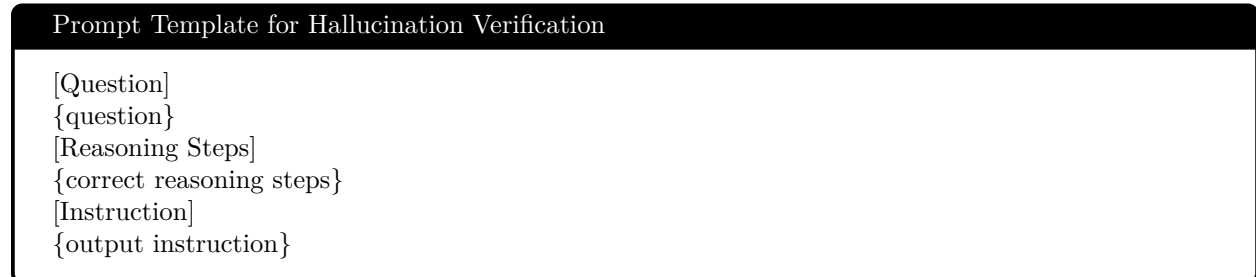

In the following, we provide the rules for judging different type of hallucination:

**Judgment Rules for Factual Inconsistency Hallucination**

The above are step-wise reasoning steps to answer the question. Please help me determine whether the last reasoning step refers factual information not mentioned before the step. All factual information should be gounded in real-world information, including:
- Known Geographic Facts: the step should include widely accepted and verifiable facts in its original format or name. For example, state the fact that "The Eiffel Tower is located in Paris.", "Mount Everest, the tallest mountain in the world, is located in the Himalayas.", etc.
- Historical Events: the step should refer historical events with correct dates or details. For example, mention that "The American Civil War ended in 1865."
- Factual Scientific Data or Statistics: the step should include correct real-world data or statistics. But, basic calculation process should not be counted as factual information.For example, a step can state that "According to the 2020 census, the population on earth is over 7.5 billion.", "There is 7 days a week.", "The pythagorean theorem is $a^2 + b^2 = c^2$.", etc.
In the output, there should be explanation whether the last reasoning step has factual information and output the facutal information first. Then, in the new line, please only output "Yes" if the last reasoning step has factual information. Otherwise, please only output "No".

**Judgment Rules for Context Inconsistency Hallucination**

The above are step-wise reasoning steps to answer the question. Please help me determine whether the last reasoning step refers question information. Referred content in the last reasoning step should be the same as it mentioned in the question. Contents indirectly related to the referred content, such as derived or concluded by the referred contents, should not be counted as question information.
In the output, there should be an explanation whether the last reasoning step refers question information, output the extact referred question information in both the last reasoning step and question first. Then, in the new line, please only output "Yes" if the last reasoning step refers question information. Otherwise, please only output "No".

**Judgment Rules for Logical Error Hallucination**

The above are step-wise reasoning steps to answer the question. Please help me determine whether the last reasoning step involves calculation processes, including mathematical calculations or formulas:
- Mathematical Calculations: the step should have at least one calculation process. The calculation processes should include numbers (3, 5, 10 etc.) or mathematical symbols (sin, cos, x, y, $\pi$, etc.), and they should be like "The sum of 45 and 15 is 60", "30*4+5=125", "sin(x)+cos(x)", etc.
- Formulas: the step should include mathematical principles, laws of physics, or other data processing operations. Formulas may be in latex format. They can be simply stated in the step and do not have equal symbols. For example, formula can be "Pi*radius$^2$", "2*Pi*radius", "[sin(x)+cos(x)]", etc.
In the output, there should be explanation whether the last reasoning step has calculation process first. Then, in the new line, please only output "Yes" if the last reasoning step has calculation process. Otherwise, please only output "No".

**Judgment Rules for Logical Inconsistency Halluciantion**

The above are step-wise reasoning steps to answer the question. Please help me determine whether the last reasoning step involves reasoning process. Referred content in the last reasoning step should be the same as it mentioned in the previous reasoning steps but not in the question. Contents indirectly related to the referred content, such as derived or concluded by the referred contents, should not be counted as previous information.

In the output, there should be an explanation whether the last reasoning step refers information in previous reasoning steps but not in the question, output the extact referred previous information in both the last reasoning step and previous step first. Then, in the new line, please only output "Yes" if the last reasoning step refers those information. Otherwise, please only output "No".

# F Prompts for Synthesizing Hallucination Dataset

We provide a system prompt template for a language model that is used to guide the larger model through the task of generating each type of hallucination. This template consists of a description of the hallucination generation task and instructions for producing a particular type of hallucination.

---

**System Prompt**

Please generate the next reasoning step to incorrectly continue the reasoning process based on a question and a series of correct reasoning steps. The next reasoning step you generate does not necessarily result in an instant final answer. And you should follow the hallucination generation instruction below to generate the next reasoning step.
{Instruction}

---

In the following, we present the instructions for generating each type of hallucination:

---

**Instruction for Generate Fabrication Hallucination**

# Instruction for Generating Fabrication Hallucination
When generating the next reasoning step, you should intentionally introduce fabrications by including facts that are unverifiable against established real-world knowledge or context information. These fabrications should be plausible within the context but should not be verifiable through any external sources. Follow these guidelines:
- Unverifiable Facts: Introduce facts that cannot be verified through established real-world knowledge. For example, mention a historical event that did not happen, or a scientific theory that does not exist.
- Fictitious Entities: Refer to people, places, or organizations that are entirely made up. For example, mention a "Dr. John Smith of the International Institute of Quantum Studies," which does not exist.,
- Imaginary Data or Statistics: Provide data or statistics that are fictional. For example, state that "according to a 2023 study by the Global Health Organization, 75% of people prefer digital books over physical ones," when no such study exists.',
# Example Guidelines
- Introduce a Fabricated Historical Event: For instance, state that "In 1875, the Grand Treaty of Lisbon established the first international postal system," even though no such treaty exists.
- Mention Nonexistent Scientific Theories or Discoveries: For example, reference "Dr. Eleanor Rigby's groundbreaking work on temporal physics, which suggests that time travel is theoretically possible," when no such work or scientist exists.
- Provide Fictitious Data or Statistics: Include statements like "A recent survey by the National Institute of Sleep Studies found that 60% of adults dream in black and white," even though such an institute or survey does not exist.
# Constraints
- Plausibility: The fabricated content should be plausible within the context but should not be verifiable.
- Consistency: The rest of the generated content should be consistent and coherent, without introducing contradictions or errors in logic.
- No Contradiction to Known Facts: Avoid contradicting widely accepted and easily verifiable facts. The fabrication should be in areas that are less likely to be immediately recognized as false.
- Maintain Context: Ensure that the fabricated information fits smoothly into the surrounding context, making it less likely to be immediately questioned.

---

---

**Instruction for Generating Factual Inconsistency Hallucination**

# Instruction for Generating Factual Inconsistency Hallucination
When generating the next reasoning step, you should intentionally introduce factual inconsistencies by including facts that can be grounded in real-world information but present contradictions. These inconsistencies should be subtle and should not be immediately obvious. Follow these guidelines:
- Contradict Known Facts: Introduce information that contradicts widely accepted and verifiable facts. For example, state that "The Eiffel Tower is located in Berlin," contradicting the well-known fact that it is in Paris.
- Inconsistent Historical Events: Reference historical events with incorrect dates or details. For example, mention that "The American Civil War ended in 1870," when it actually ended in 1865.
- Conflicting Data or Statistics: Provide data or statistics that conflict with established information. For example, state that "According to the 2020 census, the population of New York City is 2 million," when the actual population is significantly higher.
# Example Guidelines
- Contradict Known Geographic Facts: For instance, state that "Mount Everest, the tallest mountain in the world, is located in the Andes mountain range," when it is actually in the Himalayas.
- Inconsistent Historical Dates: For example, claim that "The Declaration of Independence was signed on July 4, 1800," when it was actually signed in 1776.
- Conflicting Scientific Information: Include statements like "Water boils at 110 degrees Celsius at sea level," when it actually boils at 100 degrees Celsius.
# Constraints
- Plausibility: The inconsistent content should be subtle and not immediately obvious to the reader.
- Consistency: The rest of the generated content should be consistent and coherent, without introducing contradictions or errors in logic beyond the intended inconsistencies.
- Grounded in Real-World Information: The fabricated inconsistencies should still be based on real-world information but presented inaccurately.
- Maintain Context: Ensure that the inconsistent information fits smoothly into the surrounding context, making it less likely to be immediately questioned.

---

**Instruction for Generating Instruction Inconsistency Hallucination**

# Instruction for Generating Instruction Inconsistency Hallucination
When generating the next reasoning step, you should intentionally introduce inconsistencies by not aligning the output with the specific instructions given by the user. These instruction inconsistencies should be subtle but clear enough to be identified. Follow these guidelines:
- Ignore Specific Instructions: Generate text that contradicts or disregards explicit instructions given in the prompt. For example, if asked to list developed countries in Europe, list all developed countries in the world.
- Alter the Requested Target: Change the target requested by the user. For example, if asked to list developed countries in the world, list all undeveloped countries in the world instead.
- Misinterpret the Instructions: Deliberately misinterpret the instruciton so that the output does not respond directly to the user's request. For example, if asked for "Japan's capital city", answer "Japan's largest city is Tokyo", even though Tokyo is the largest city in Japan.
# Constraints
- Faithful: You cannot fabricate something that doesn't appear in the context.
- Coherence: The rest of the generated content should remain coherent and logical, without introducing contradictions or errors beyond the intended inconsistencies.
- Contextual Fit: Ensure that despite the inconsistency, the response still fits smoothly within the broader context of the conversation or text, making it less likely to be immediately questioned.

**Instruction for Generating Context Inconsistency Hallucination**

# Instruction for Generating Context Inconsistency Hallucination
When generating the next reasoning step, you should introduce inconsistencies by intentionally modifying information to contradict the user's provided contextual information. These context inconsistencies should be subtle but clear enough to be identified. Follow these guidelines:
- Contradict Provided Facts: Introduce information that directly contradicts the facts given in the user's prompt. For example, if the user states that "Bob was born in England," you may contradict it by stating that "Bob was born in France."
- Alter Specific Details or Data: Change specific details or data provided by the user. For example, if the user mentions that "Bob has three books and two pens in his backpack," you might alter it by stating that "Bob has two books and four pens in his backpack."
- Misattribute Quotes or Data: Attribute quotes or data to the wrong source. For example, if the user states that "Bob likes apples while Jane likes bananas." you might contradict it by stating "Jane likes apples" or "Bob likes bananas".
# Constraints
- Subtlety: The context inconsistencies should be subtle and not immediately obvious to the reader.
- Coherence: The rest of the generated content should remain coherent and logical, without introducing contradictions or errors beyond the intended inconsistencies.
- Contextual Fit: Ensure that the inconsistent information fits smoothly within the broader context of the conversation or text, making it less likely to be immediately questioned.

**Instruction for Generating Logical Inconsistency Hallucination**

# Instruction for Generating Logical Inconsistency Hallucination
When generating the next reasoning step, you should introduce logical inconsistencies by incorrectly referring to or copying content from previous reasoning steps. These logical inconsistencies should be subtle but clear enough to be identified. Follow these guidelines:
- Incorrect Reference: Refer to a previous reasoning step incorrectly, such as misinterpreting or misrepresenting the calculations or conclusions. For example, if a previous step states "Bob is an undergraduate," you may incorrectly refer back to this by stating "Since Bob is a graduate..."
- Copying Errors: Copy content from a previous reasoning step but alter it in a way that introduces an error, such as changing numbers or relationships. For example, if the reasoning involves steps for calculating a total cost and one step states "Item A costs 5 * 2 =10," you might incorrectly copy this as "Since item A costs 5 * 3 =15..." in the next step.
- Make logical leaps or conclusions that do not follow from the previous steps, leading to an incorrect answer.
# Constraints
- Subtlety: The logical inconsistencies should be subtle and not immediately obvious to the reader.
- Coherence: The rest of the generated content should remain coherent and logical, without introducing contradictions or errors beyond the intended inconsistencies.
- Contextual Fit: Ensure that the inconsistent information fits smoothly within the broader context of the conversation or text, making it less likely to be immediately questioned.

**Instruction for Generating Calculation Error Hallucination**

# Instruction for Generating Calculation Error Hallucination
When generating the next reasoning step, you should intentionally introduce calculation error by including incorrect numerical calculations or data processing. These errors should be subtle but clear enough to be identified. Follow these guidelines:
- Perform Erroneous Mathematical Calculations: Make intentional mistakes in mathematical calculations. For example, state that "The sum of 45 and 15 is 70", when it is actually 60.
- Include Incorrect Data Processing: Misapply mathematical principles, laws of physics, or other data processing operations. For example, when asked to calculate the area of a circular, compute the perimeter formula 2*Pi*radius instead of the area formula Pi*radius$^2$.
- Generates responses with unsupported claims, including numerical assertions that have no basis in the provided context or input.
# Constraints
- The values you use must be consistent with the context given, but the final calculation should be intentionally miscalculated.
- You must not fabricate what does not appear in the context or contradict widely accepted and easily verifiable facts.
- Ensure that despite the errors, the response still fits smoothly within the broader context of the conversation or text.

**Instruction for Generating Logical Error Hallucination**

# Instruction for Generating Logical Error Hallucination
When generating the next reasoning step, you should intentionally introduce logical error by including flawed logical reasoning or incorrect inferences. These errors should be subtle but clear enough to be identified. Follow these guidelines:
- Causal Misattribution: Incorrectly identify the cause of an event or outcome. For example, conclude that "Because it rained yesterday, that's why the football team won today's match," without considering other relevant factors.
- Overgeneralization: Apply a rule or pattern more broadly than it should be. For instance, generalize that "all mammals fly" based on the fact that bats are flying mammals.
- Generate responses with unsupported claims, including assertions that do not logically follow from the premises provided.
# Constraints
- The information you refer to must be consistent with the information provided in the previous reasoning steps and context, but the final conclusion should be intentionally and logically flawed.
- You must not fabricate what does not appear in the context or contradict widely accepted and easily verifiable facts.

# G   Hallucination Demonstration Examples

We provide demonstrations for generating each type of hallucination. Each demonstration includes two examples of an injecting hallucination, along with an explanation of how it is produced.

---

**Demonstrations for Fabrication Hallucination**

[Question]
What are the primary components of DNA?
[Correct Reasoning Steps]
Step 1: DNA is structured as a double helix composed of nucleotides.
Step 2: Each nucleotide consists of a sugar (deoxyribose), a phosphate group, and a nitrogenous base.
Step 3: The four nitrogenous bases are adenine (A), thymine (T), cytosine (C), and guanine (G).
{output format}
[Explanation]
The user is asking about the primary components of DNA. The correct approach is to describe the structure of DNA and its components, including the nucleotides and the four nitrogenous bases. The Next Reasoning Step here introducesFabrication Hallucination by mentioning a "recent study by the Molecular Genetics Institute in Zurich" that identified a fifth nitrogenous base, "neomine (N)," which does not exist. This reasoning step remains coherent and logical, correctly describing the structure of DNA and its primary components, but introducing a fictitious base and study that is not supported by established scientific knowledge.
[Next Reasoning Step with Fabrication Hallucination]
Step 4: Additionally, a recent study by the Molecular Genetics Institute in Zurich has identified a rare fifth base, neomine (N), which appears to play a role in DNA repair mechanisms.
[Question]
What are the main causes of the decline of the Roman Empire?
[Correct Reasoning Steps]
Step 1: One of the major causes of the decline of the Roman Empire was economic trouble, which included heavy taxation and inflation.
Step 2: Additionally, the overreliance on slave labor created economic stagnation.
Step 3: Military overspending further strained the empire's resources.
{output format}
[Explanation] The user is asking about the main causes of the decline of the Roman Empire. The correct approach is to identify a combination of internal weaknesses and external pressures that contributed to the empire's fall, along with key events that marked its decline. The Next Reasoning Step here introduces Fabrication Hallucination by mentioning the "Great Plague of 330 AD," which did not actually occur, and the Roman historian "Lucius Marcellus," who is not a verifiable historical figure. This reasoning step remains coherent and logical, offering a plausible explanation for the decline of the Roman Empire, but introducing a fictitious event and historian that can not verified by known historical facts.
[Next Reasoning Step with Fabrication Hallucination]
Step 4: A significant but lesser-known factor was the Great Plague of 330 AD, which decimated the population and weakened the military. This pandemic, documented by the Roman historian Lucius Marcellus, is believed to have killed nearly a quarter of the empire's population.

---

**Demonstrations for Context Inconsistency Hallucination**

[Question]
Ann's favorite store was having a summer clearance. For $75 she bought 5 pairs of shorts for $7 each and 2 pairs of shoes for $10 each. She also bought 4 tops, all at the same price. How much did each top cost?
[Correct Reasoning Steps]
Step 1: She bought 5 shorts at $7 each so 5*7=$35
Step 2: She bought 2 pair of shoes at $10 each so 2*10=$20
Step 3: The shorts and shoes cost her 35+20 = $55
Step 4: We know she spent 75 total and the shorts and shoes cost $55 which left a difference of 75-55 = $20
{output format}
[Explanation]
The user is asking about how much each top cost based on the total amount spent. The correct approach is to calculate the total cost of shorts and shoes first, then determine the remaining amount spent on tops, and finally find the cost per top by dividing the remaining amount by the number of tops. The Next Reasoning Step here introduces Context Inconsistency Hallucination by incorrectly stating that Ann bought 5 tops instead of the 4 tops mentioned in the user's provided context, which creates a context inconsistency by altering the specific detail about the number of tops. This reasoning step maintains coherent and logical in its structure, providing a plausible answer for the price of each top, but introduces a subtle context inconsistency in the number of tops that affects the final answer.
[Next Reasoning Step with Context Inconsistency Hallucination]
Step 5: Since she bought 5 tops for a total of $20, so each top cost 20/5 = $4
[Question]
There are 16 dolphins in a water park. One quarter of them are gray dolphins, and there are twice as many white dolphins as gray dolphins. How many white dolphins are in the water park?
[Correct Reasoning Steps]
Step 1: We know one quarter of the dolphins are gray, we calculate this by dividing the total number of dolphins by 4, 16/4 = 4
{output format}
[Explanation]
The user is asking about the number of white dolphins in the water park, given the total number of dolphins and the proportion of gray dolphins. The correct approach is to determine the number of gray dolphins first, then use the given ratio to find the number of white dolphins. The Next Reasoning Step here introduces Context Inconsistency Hallucination by incorrectly stating that there are twice as many gray dolphins as white dolphins, instead of correctly stating that there are twice as many white dolphins as gray dolphins. This creates a context inconsistency by altering the specific relationship between the number of gray and white dolphins. This reasoning step remains coherent and logical in its structure, providing a plausible method to find the number of white dolphins, but introduces a subtle context inconsistency in the relationship between the number of gray and white dolphins that contradict provided facts.
[Next Reasoning Step with Context Inconsistency Hallucination]
Step 2: Since there are twice as many grey dolphins as white dolphins, we divide the number of gray dolphins by 2 to get the number of white dolphins: 4/2=2.

**Demonstrations for Instruction Inconsistency Hallucination**

[Question]
Adam bought 3 kilograms of nuts and 2.5 kilograms of dried fruits at a store. One kilogram of nuts costs $12 and one kilogram of dried fruit costs $8. How much did his purchases cost?
[Correct Reasoning Steps]
Step 1: For the nuts Adam paid 3 * 12 =36.
Step 2: And for dried fruits Adam paid 2.5 * 8 =20.
{output format}
[Explanation]
The user is asking for the total cost of Adam's purchases. The correct next reasoning step should add the costs of the nuts and dried fruits to find the total cost. The Next Reasoning Step here introduces Instruction Inconsistency Hallucination by calculating the average cost of the purchases instead of finding the total cost, altering the requested target. Despite the inconsistency, this reasoning step introduces no contradictions or errors in logic, and still fits smoothly within the broader context of the conversation.
[Next Reasoning Step with Instruction Inconsistency Hallucination]
Step 3: To find the average cost of Adam's purchases, we can add the cost of nuts and dried fruits and divide by 2: ($36 + $20) / 2 = $28.
[Question]
Abigail is trying a new recipe for a cold drink. It uses 14 of a cup of iced tea and 1 and 14 of a cup of lemonade to make one drink. If she fills a pitcher with 18 total cups of this drink, how many cups of lemonade are in the pitcher?
[Correct Reasoning Steps]
Step 1: Each drink uses 1.5 cups because 14 cup + 1 and 14 cup = 1.5 cups
Step 2: The pitcher contains 12 total drinks because 18  1.5 = 12
{output format}
[Explanation]
The user is asking the number of cups of lemonade in the pitcher. The next correct reasoning step should calculate the total cups of lemonade by multiplying the number of drinks by the amount of lemonade per drink. The Next Reasoning Step here introduces Instruction Inconsistency Hallucination by suddenly changing the unit of measurement from cups to ounces, ignoring the specific instruction to find the number of cups. Despite the inconsistency, this reasoning step introduces no contradictions or errors in logic, and still fits smoothly within the broader context of the conversation.
[Next Reasoning Step with Instruction Inconsistency Hallucination]
Step 3: Since each drink uses 1 and 1/4 cups of lemonade, and there are 8 ounces in a cup, the total ounces of lemonade in the pitcher are 12 * (1 and 1/4) * 8 = 96 ounces.

**Demonstrations for Logical Inconsistency Hallucination**

[Question]
Annie, Bob, and Cindy each got some candy. Annie has 6 candies, Bob has 2 candies more than half of Annie's candies, and Cindy has 2 candies less than twice Bob's candies. Which of the three of them has the least amount of candy?
[Correct Reasoning Steps]
Step 1: Annie has 6 candies.
Step 2: Bob has 2 candies more than half of Annie's candies. Half of Annie's candies is ( 6 / 2 = 3 ). So, Bob has ( 3 + 2 = 5 ) candies.
Step 3: Cindy has 2 candies less than twice Bob's candies. Twice Bob's candies is ( 2 * 5 = 10 ). So, Cindy has ( 10 - 2 = 8 ) candies.

{output format}
[Explanation] The user is asking which of Annie, Bob, and Cindy has the least amount of candy. The correct approach is to calculate the number of candies each person has and then compare these amounts to determine who has the least. According to the previous steps: 1. Annie has 6 candies; 2. Bob has 5 candies; 3. Cindy has 8 candies. The Next Reasoning Step here introduces Logical Inconsistency Hallucination by incorrectly concluding that Annie has the least amount of candy, whereas the correct conclusion should be that Bob has the least amount of candy with 5 candies. This creates a logical inconsistency by failing to accurately reference the correct comparative amounts of candies, contradicting the previous reasoning steps.
[Next Reasoning Step with Logical Inconsistency Hallucination]
Step 4: Since Annie only has 6 candies, Anne has the least amount of candy.
[Question]
Annie, Bob and Cindy each buy personal pan pizzas cut into 4 pieces. If Bob eat 50% of his pizzas and Ann and Cindy eat 75% of the pizzas, how many pizza pieces are left uneaten?
[Correct Reasoning Steps]
Step 1: In total, there are 3 * 4 = 12 pizza pieces. Step 2: Bob eats 4 * 50% = 2 pieces. Step 3: Annie and Cindy eat 2 * 4 * 75% = 6 pieces. Step 4: The three of them eat 2 + 6 = 8 pieces.
{output format]}
[Explanation]
The user is asking how many pizza pieces are left uneaten after Annie, Bob and Cindy each eat a portion of their pizzas. The correct approach is to calculate the total number of pizza pieces, determine how many pieces each person eats, and then find the remaining uneaten pieces. According to the previous steps: 1. In total, there are 12 pizza pieces; 2. Bob eats 2 pieces; 3. Annie and Cindy together eat 6 pieces; 4. Therefore, the three of them eat 2 + 6 = 8 pieces. The Next Reasoning Step here introduces Logical Inconsistency Hallucination by incorrectly copying that 10 pieces of pizza were eaten and by incorrectly referencing the total number of pizza pieces as 16, whereas the correct calculation should be based on the total number of 12 pizza pieces and the remaining uneaten pieces should be 12 - 8 = 4. This creates a logical inconsistency by incorrectly referencing the number of eaten pieces as 10 and the total number of pizza pieces as 16, contradicting the previous reasoning steps.
[Next Reasoning Step with Logical Inconsistency Hallucination]
Step 5: Since 10 pizza pieces were eaten, there are 16 - 10 = 6 pizza pieces uneaten.

## Demonstrations for Calculation Error Hallucination

[Question]
Abigail is trying a new recipe for a cold drink. It uses 0.25 of a cup of iced tea and 1.25 of a cup of lemonade to make one drink. If she fills a pitcher with 18 total cups of this drink, how many cups of lemonade are in the pitcher?
[Correct Reasoning Steps]
Step 1: Each drink uses 1.5 cups because 0.25 cup + 1 and 1.25 cup = 1.5 cups
Step 2: The pitcher contains 12 total drinks because 18 / 1.5 = 12
{output format}
[Explanation]
The user is asking for the total cups of lemonade in the pitcher. The correct calculation involves finding the total number of drinks and then multiplying it by the amount of lemonade per drink. The Next Reasoning Step here introduces Calculation Error Hallucination by incorrectly calculating the total cups of lemonade as 12 * 1.25 = 17 cups when it is actually 12 * 1.25 = 15 cups. Despite the incorrect calculation, this reasoning step correctly identifies the values to be multiplied, and still fits smoothly within the broader context of the conversation.
[Next Reasoning Step with Calculation Error Hallucination]
Step 3: Since each drink uses 1.25 cups of lemonade, multiply the total drinks by the amount of lemonade per drink: 12 * 1.25 = 17 cups of lemonade.
[Question]
Queenie earns \$150 a day as a part-time clerk. She earns an additional \$5 per hour as overtime pay. How much will Queenie receive for working 5 days with 4 hours overtime?
[Correct Reasoning Steps]
Step 1: Queenie will earn \$150 * 5 = \$750 for working 5 days Step 2: She will receive an additional \$5 * 4 = \$20 for overtime pay.
{output format}
[Explanation]
The user is asking for Queenie's total earnings for working 5 days with 4 hours of overtime. The correct calculation involves adding her daily earnings to her overtime pay. The Next Reasoning Step here introduces Calculation Error Hallucination by incorrectly adding \$750 and \$20 as \$7800 when it is actually \$770. Despite the incorrect calculation, this reasoning step correctly identifies the values to be added, and still fits smoothly within the broader context of the conversation.
[Next Reasoning Step with Calculation Error Hallucination]
Step 3: Then, add her daily earnings and overtime pay to get her total earnings: \$750 + \$20 = \$7800.
[Question]
What is the volume of a cylinder with a radius of 3 units and a height of 5 units?
[Correct Reasoning Steps]
Step 1: The volume of a cylinder is calculated using the formula Volume $= \pi \times \text{radius}^2 \times \text{height}$.
Step 2: For a cylinder with a radius of 3 units and a height of 5 units, first calculate $\pi \times 3^2 = 9\pi$.
{output format}
[Explanation]
The user is asking for the volume of a cylinder. The correct formula involves multiplying $\pi$ by the square of the radius and then by the height. Given the radius is 3 units and the height is 5 units, the volume should be calculated as$\pi \times 3^2 \times 5 = 45\pi$. The Next Reasoning Step here introduces Calculation Error Hallucination by incorrectly calculating $9\pi$ multiplied by 5 as $18\pi$ when it is actually $45\pi$. Although the final result is miscalculated, this reasoning step correctly identifies the values to be multiplied, and still fits smoothly within the broader context of the conversation.
[Next Reasoning Step with Calculation Error Hallucination]
Step 3: Then multiply by 5, and the volume is $9\pi \times 5 = 18\pi$ cubic units.

---

**Demonstrations for Logical Error Hallucination**

[Question]
Please answer the question in detail and accurately based on the information provided in the following sentences.
Question: Kevin is observing the sky on a clear night. With the unaided eye he is able to see Venus, Mars, Jupiter, and Saturn. Why would Venus appear to be brighter than the other planets?
Information: sent1: more light gets reflected on highly reflective things sent2: venus is covered in highly reflective clouds sent3: as the light reflected off of an object increases , the object will appear to be brighte
[Correct Reasoning Steps]
Step 1: More light gets reflected on highly reflective things.
Step 2: Venus is covered in highly reflective clouds.
{output format}
[Explanation]
The user is asking why Venus appears brighter than other planets in the night sky. The correct reasoning involves recognizing that Venus has highly reflective clouds, which contribute to its brightness. However, other factors like its relative proximity to Earth and its position in the sky also play significant roles. The Next Reasoning Step here introduces a Logical Error Hallucination by incorrectly inferring that because Venus is highly reflective, it must also be the closest planet to Earth. This is a flawed causality because the reflectivity of Venus's clouds does not determine its distance from Earth, which leads to an incorrect conclusion about Venus's position relative to Earth and its apparent brightness.
[Next Reasoning Step with Calculation Error Hallucination]
Step 3: Since Venus is covered in highly reflective clouds, it reflects more light than any other planet, making it the closest planet to Earth and therefore the brightest in the night sky.
[Question] Please answer the question in detail and accurately based on the information provided in the following sentences.
Question: Which form of energy is needed to change water from a liquid to a gas?
Information: sent1: gas is a kind of state of matter sent2: water is a kind of substance sent3: liquid is a kind of state of matter sent4: heat energy can change the state of matter
[Correct Reasoning Steps]
Step 1: Gas is a kind of state of matter.
Step 2: Liquid is a kind of state of matter.
Step 3: Heat energy can change the state of matter.
{output format}
[Explanation]
The user is asking which form of energy is needed to change water from a liquid to a gas. The correct reasoning involves understanding that heat energy is required to change the state of matter, specifically from liquid to gas. However, the Next Reasoning Step here introduces a Logical Error Hallucination by incorrectly inferring that since heat energy can change the state of matter, any form of energy can change water from liquid to gas. This is a inductive reasoning error because while heat energy specifically is required for this phase change, not all forms of energy are applicable. The overgeneralization in this step leads to an incorrect conclusion about the types of energy that can achieve this state change.
[Next Reasoning Step with Logical Error Hallucination]
Step 4: Since heat energy can change the state of water from a liquid to a gas, any form of energy can be used to change water from a liquid to a gas.

