# OpenReview forum: "FG-PRM: Fine-grained Hallucination Detection and Mitigation in Language Model Mathematical Reasoning"
_TMLR — Rejected by TMLR_

### Review · Reviewer_d21a · 2024-12-02

**Summary Of Contributions:**

The authors claim three main contributions in the field of hallucination mitigation for ground-truth (and in particular mathematics) problems. First, they introduce a comprehensive taxonomy that categorizes the common hallucinations in mathematical reasoning task into six types: fabrication, factual inconsistency, context inconsistency, instruction inconsistency, logical inconsistency, and logical error. They then propose FG-PRM (Fine-Grained Process Reward Model), an augmented model designed to detect and mitigate hallucinations in a fine- grained, step-level manner. The authors automate the synthesis (and categorization) of fine-grained hallucination data using LLMs. By injecting hallucinations into reasoning steps of correct solutions, they create a diverse and balanced synthetic dataset for training FG-PRM, which consists of six specialized Process Reward Models (PRMs), each tailored to detect a specific hallucination type. The authors then conduct experiments which show improvement over naive baselines in some hallucination categories.

**Audience:**

Yes

**Broader Impact Concerns:**

No broader impact concerns.

**Claims And Evidence:**

No

**Requested Changes:**

* The authors claim, as a contribution, to have devised a comprehensive hallucination taxonomy that categorizes common errors in mathematical reasoning tasks into six distinct types, building on the long-standing existing two-part taxonomy (intrinsic, manipulating the information present in the input document, and extrinsic, adding information not directly inferable from the input document).

Q1. Since a user's request is itself provided contextual information, wouldn't Instruction Inconsistency, as currently defined, be a subset of Context Inconsistency? The example you give in Table 4 is a failure to reproduce the instruction correctly. Does Instruction Inconsistency perhaps only refer to a failure to correctly reproduce the instructions?

Q2. Wouldn't Logical Inconsistency be a form of extrinsic rather than intrinsic hallucination, since the mistakes derive from the model's own output? Your example in Table 4 illustrates this well.

Q3. How is Logical Error, in which conclusions that do not follow from the provided premises, distinct from Logical Inconsistency, inconsistency between steps and final answer? Also, in the example given in Table 4, both are extrinsic errors, although I don't think this would always be the case.

**S1.** Based on the above questions, I suggest that the authors consider rephrasing and perhaps recategorizing some of the elements in their taxonomy.

* The claim in your paper that "we demonstrate that our FG-PRM demonstrates superior performance in the hallucination detection task compared to ChatGPT-3.5 and Claude-3" is not supported by the evidence.  FG-PRM is not the best-performing model, on average, in either setting of Table 1.

**S2.** This claim should be reworded.

**Strengths And Weaknesses:**

The subject matter, hallucination mitigation, is timely, and will certainly be of interest to some members of the TMLR community.

The authors are to be commended for the outstanding visual presentation of this work. The figures, tables and color coding are extremely helpful, and make the paper more enjoyable to review.

---

> ### Author Response · Authors · 2024-12-05
> **Response to Questions**
>
> Thank you for the comprehensive and insightful feedback! We appreciate your effort in reviewing our paper and are committed to addressing your concerns in the following response.
>
> > 1. **Since a user's request is itself provided contextual information, wouldn't Instruction Inconsistency, as currently defined, be a subset of Context Inconsistency? The example you give in Table 4 is a failure to reproduce the instruction correctly. Does Instruction Inconsistency perhaps only refer to a failure to reproduce the instructions correctly?**
>
> Context Inconsistency differs from Instruction Inconsistency. **Context Inconsistency** occurs when the reasoning process incorrectly refers to **information** from the context. ***Instruction Inconsistency*** arises when the reasoning process fails to adhere to ***instructions***. In Table 4, the example of Instruction Inconsistency involves outputting a decimal number instead of the required fraction, contrary to the ***instruction*** to "represent ... as a fraction", which serves not as information used for reasoning but as a ***directive for the output format***. Conversely, the example of Context Inconsistency demonstrates an error where the reasoning process misinterprets or misuses **information** from the context. **The modifications for this question and hallucination definition have been marked as red text in our manuscript, as well as for the following three questions.**
>
> | Hallucination Sub-Type    | User Input                                                                                 | Model Output                                                                                                                     | Explanation                                                                                                                                                                                                   |
> | ------------------------- | ------------------------------------------------------------------------------------------ | -------------------------------------------------------------------------------------------------------------------------------- | ------------------------------------------------------------------------------------------------------------------------------------------------------------------------------------------------------------- |
> | Instruction Inconsistency | Please calculate the quotient of 25 divided by 100 and represent the result as a fraction. | The quotient of 25 divided by 100 is 0.25.                                                                                       | Although the model correctly calculates the result 0.25, it fails to follow the "as a fraction" **instruction** in the user input.                                                                            |
> | Context Inconsistency     | Given that I have 5 apples and I **give 2** to my friend, how many apples do I have left?  | Considering the information provided, if you initially had 5 apples and **gave 3** to your friend, you would have 2 apples left. | In the bold reasoning step, the model incorrectly refers to the number of apples (the **information**). The user stated that they gave 2 apples to their friend, but the model incorrectly refers to it as 3. |
>
> > 2. **Wouldn't Logical Inconsistency be a form of extrinsic rather than intrinsic hallucination, since the mistakes derive from the model's own output? Your example in Table 4 illustrates this well.**
>
> No. We classify hallucinations as intrinsic and extrinsic based on whether they can be verified by the input information or the context previously generated by LLMs. Logical Inconsistency refers to hallucinations arising in a reasoning step that incorrectly references details from earlier reasoning steps. Since all relevant information is contained within the reasoning process and **can be verified internally without requiring external knowledge or tools**, Logical Inconsistency is categorized as an intrinsic hallucination.

---

> ### Author Response · Authors · 2024-12-05
> **Response to Questions Cont.**
>
> 3. **How is Logical Error, in which conclusions that do not follow from the provided premises, distinct from Logical Inconsistency, inconsistency between steps and final answer? Also, in the example given in Table 4, both are extrinsic errors, although I don't think this would always be the case.**
>
>
> Logical Error happens in a single reasoning step, while Logical Inconsistency refers to hallucinations between the current and previous steps. In Table 4, Logical Error is categorized as an extrinsic hallucination, whereas Logical Inconsistency is classified as an intrinsic hallucination.
>
> We appreciate your feedback regarding the need for more clarity in the Logical Error example in Table 4 and have updated the example in our manuscript. The updated example replaces the current step with: “So, the perimeter would be 2x(4+3)=16. So the answer is 16”. This revised example explicitly illustrates a Logical Error hallucination, as the calculation operation in the current step is incorrect. The revised sample in Table 4 is:
>
> | Hallucination Sub-Type | User Input                                                                                             | Model Output                                                                                                                                                                                          | Explanation                                                                                                                                                                                                       |
> | ---------------------- | ------------------------------------------------------------------------------------------------------ | ----------------------------------------------------------------------------------------------------------------------------------------------------------------------------------------------------- | ----------------------------------------------------------------------------------------------------------------------------------------------------------------------------------------------------------------- |
> | Logical Error          | If a rectangle has a length of 4 units and a width of 3 units, what is the perimeter of the rectangle? | The perimeter formula of a rectangle is: Perimeter = 2 × (length + width). The length is 4 units and the width is 3 units. **So, the perimeter would be 2 × (4 + 3)=16.** So, the answer is 16 units. | In the bold step, the model incorrectly calculates 2x(4+3) as 16, which should be 14. The error should be verified by external tools like calculators, which match the definition of Logical Error hallucination. |
>
> > 4. **The claim in your paper that "we demonstrate that our FG-PRM demonstrates superior performance in the hallucination detection task compared to ChatGPT-3.5 and Claude-3" is not supported by the evidence. FG-PRM is not the best-performing model, on average, in either setting of Table 1. This claim should be reworded.**
>
> Thanks for pointing out the issue. We will revise the claim as follows:
>
> “We demonstrate that our FG-PRM surpasses ChatGPT-3.5 and Claude-3 in the hallucination detection task for most hallucination types, achieving over 5% higher F1 scores in these cases.”
>
> This revised claim is supported by Table 1 in our manuscript, where FG-PRM outperforms the other two models in 4 out of 6 hallucination types. The two models outperform our FG-PRM on Factual Inconsistency and Fabrication, mainly because the two commercial models are much larger than our FG-PRM and store a greater amount of fact-based knowledge.
>
> Moreover, we conduct a new experiment on PRM on the hallucination detection task in Table 1. The results are as follows:
>
> Synthetic Reasoning Chain:
>
> | Detector | CI    | LI    | II    | LE    | FI    | FA    | Avg.  |
> | -------- | ----- | ----- | ----- | ----- | ----- | ----- | ----- |
> | PRM      | 0.399 | 0.455 | 0.467 | 0.402 | 0.358 | 0.565 | 0.441 |
>
> Human-annotated Reasoning Chain:
>
> | Detector | CI    | LI    | II    | LE    | FI    | FA    | Avg.  |
> | -------- | ----- | ----- | ----- | ----- | ----- | ----- | ----- |
> | PRM      | 0.394 | 0.493 | 0.484 | 0.357 | 0.403 | 0.435 | 0.428 |
>
> The results show a similar trend as FG-PRM, but PRM performs worse than FG-PRM.

---

### Review · Reviewer_y8A3 · 2024-12-09

**Summary Of Contributions:**

The authors introduce a new taxonomy for (LLMs) hallucinations in mathematical reasoning. They propose a method to generate data according to this taxonomy. Moreover, they propose a new framework that extends process reward models (PRMs) with more fine-grained (i.e. less distant) supervision (FG-PRM).

**Audience:**

Yes

**Claims And Evidence:**

No

**Requested Changes:**

I reckon the paper in this state is not ready for publication. However, I think there are good ideas and results that are worth to be published. For me to consider it good enough, the concerns in the weaknesses section should be all carefully addressed. Moreover, the following points should be fixed:
- In Section 6, in the OOD paragraph “As detailed in Figure 4, the gap in CG-H (+0.3) is more significant than in FG-H (+0.1).” the numbers should be 0.03 and 0.01
- In Figure 3, if I correctly interpret y-axis to be accuracy (as you state in 5.3 by saying “Longformer and Llama3-8B achieve 94% and 58% accuracy on GSM8k and MATH”) then write “accuracy” as a label, and not “percentage”
- In 4.2.2, “where the [sep] token represents the classification output at each reasoning step to indicate whether the following step y i contains the hallucination type t.” Instead of “following” it should be “previous”, I suppose
- You should describe better what the self-consistency is. It sounds important to understand how you find the baselines, and right now it is not clear to me why such a computation is a good baseline. Maybe an example / intuitive explanation can help.
- End of 5.1 you say “In inference processes, we sum all results as our final results”. This is not clear to me. Do you refer to Eq. 6? If yes then refer to it, or drop the sentence completely since it is already explained above.
- In 5.1, Human-annotated Data, this sentence is not clear to me “Moreover, each type of hallucination has 50 human-annotated reasoning chains, along with the corresponding hallucinations.”
- Spotted typos:
    - In 5.3 “verifiers become more *evidence* in the more complex MATH dataset”
    - In sec 8 “[…] we propose FG-PRM framework,introduces […]”

Finally, I think the paper can improve a lot with a careful re-writing of section 5, 6 and 8 aimed at improve clarity. I have some questions that might help me understand some details / improve the work:
- In the dataset you generate with specific hallucination types. What happens if more hallucinations are present at the same time (in a single step)? Is it possible or do you have a way of guaranteeing a single hallucination per step?
- In the hallucination detection task, would a random classifier have 50% accuracy? If yes, how do the different methods compare to it? If not, what is the hardness of the task then? Could be good to explain this better and include the accuracy also, if you think it makes the discussion clearer.
- About the datasets, if I correctly understand, you train the 6 models that compose a FG-PRM separately, so you have in practice 6 separate datasets, one per hallucination type?
- In section 4.2.1, Step 1, you say “To achieve this, we have developed a set of tailored rules for the Llama3-70B”. These rules are “prompts”, right? I would call them “prompts” then, and not “rules”, otherwise it sounds more formal than what it is.

**Strengths And Weaknesses:**

### STRENGTHS
- The problem is very relevant and important to address
- The taxonomy seems well studied and well presented
- Contributions are clearly stated (even if a bit over-sold, see weaknesses)
- Overall the exposition is clear and contains good examples to help understanding (I do appreciate the colouring of the different terms to help distinguish them)

### WEAKNESSES
The main problem is that there are several mistakes and imprecisions that undermine the soundness of the work.
Specifically,
- There are wrongly-highlighted best results in the tables. In Table 1, PRM is the best for LE (Synth), Claude is the best for FA (Synth), ChatGPT is equally good as FG-PRM for LE (Human-annotated). In Table 5 and 6 in appendix also ChatGPT is the best for LE and LI, respectively.
- There are serious issues with reproducibility. The model used is not described at all, not in terms of architecture nor in terms of parameters required. The only reference is in the experimental results where they say that FG-PRM has less parameters and LLMs (to explain the results). From what I understand FG-PRM is “just” a collection of 6 PRMs, so one may assume they used an architecture similar to previous work, but this should be stated clearly. There is no description of hyper parameters. No mention of repeated runs, no statistical features (e.g. std error) in the results (only exception in caption of Table 2 where they say that “most improvements […] have p <0.05”, but “most” is not scientific, which ones are significant exactly?). No mention of the hardware used to run the experiments.
- On the same note, the datasets used should be better described (e.g. In Table 1, Synthetic comes from GSM8k? And Human-annotated from MATH? And what about the Math-Shepherd? There should be a clear definition for each dataset with a unique name that then is used across the various experiments). Section 5 lacks clarity in general, but in particular it is very confusing what datasets are used for what. Right after Table 1 it is mentioned 700/100 samples from GSM8k and MATH for train/test. But then these numbers become 12,000, and it is not clear how (especially because they say the split of train/test is 95/5%). Also in the OOD experiments, different datasets are used but they are not explained at all.
- The two main contribution of the paper are, in my opinion, the introduction of the taxonomy and the procedure to augment the datasets with the hallucination types introduced in the taxonomy. The problem is that both are not very well supported by experiments or explanation of any kind.
    - For the introduction of the taxonomy, despite I agree it looks like it makes sense, it would be interesting to see experiments or a more in-depth explanation on why this is a sound choice. The only reference to this I can find is in the experimental section, where they say that “These solutions are then manually annotated by three graduate students using the taxonomy of hallucination types proposed in Section 2. The mutual agreement among annotators is 79%”. This should be better highlighted an expanded as a way to indicate that the selected taxonomy is a good one for this domain.
    - For what concerns the dataset augmentation with hallucinations, since they use LLMs to generate new data, one would immediately wonder how good these generations are. Especially in light of the discussion on the manually annotated examples. If three students agree on 79% of the cases, how does this translates to the LLM generation of these annotations?
- Regarding the experiment on hallucination detection, I think the authors are overstating the results. I do think that there is value in (semi-)negative results, but it is not appropriate to present them as significant improvements, when it is not the case. I recommend rephrasing, e.g. already in the abstract they state “Our experiments show that FG-PRM outperforms ChatGPT-3.5 and Claude-3 on fine-grained hallucination detection”, this is true only for some types of hallucinations, and not even the majority of them; another example is in 5.2 where they say “[…] FG-PRM has effectively learned the patterns of these hallucinations and is capable of detecting them accurately”, given the results I think this sounds too strongly stated. On the other hand, I find it interesting that a more direct supervision does not necessarily lead to better performance.

---

> ### Author Response · Authors · 2024-12-24
> **Reply to Questions**
>
> Thank you for your thoughtful and constructive feedback. We have addressed your concerns and responded to your questions in detail below. All updates and revisions have been highlighted in red in the revised PDF for clarity.
>
> > 1.  There are wrongly-highlighted best results in the tables.
>
> We have updated them in the latest version of PDF.
>
> > 2.  The model used is not described at all, not in terms of architecture nor in terms of parameters required. The only reference is in the experimental results where they say that FG-PRM has less parameters and LLMs (to explain the results). From what I understand FG-PRM is “just” a collection of 6 PRMs, so one may assume they used an architecture similar to previous work, but this should be stated clearly.
>
> In section 5.1, we mentioned models, “We apply the LongFormer-base-4096 (Beltagy et al., 2020) and Llama-3-8B (Dubey et al., 2024) as our base models.” We add an additional description in the PDF, “We do not modify their main structure in our experiments. In detail, to build a reward model, we replace the output layer with an MLP layer to predict binary hallucination labels.”
>
> > 3.  No mention of repeated runs, no statistical features (e.g. std error) in the results (only exception in caption of Table 2 where they say that “most improvements […] have p <0.05”, but “most” is not scientific, which ones are significant exactly?)
>
> We have updated Table 2 in the PDF for more clarification. About repeated runs, we have mentioned it in the caption of Table 2, “Each result is the mean of results from 3 groups of sampling results”. Moreover, in Table 2, only two data points do not pass the significant test with p<0.05. We have added a mark after each data point. All significant tests are based on the results of the self-consistency method.
>
> > 4.  No mention of the hardware used to run the experiments.
>
> Thanks for your suggestion. We add a related description in section 5.1 in the PDF, “We use 4 Nvidia A100 80G GPUs for all of our experiments.”
>
> > 5.  the datasets used should be better described (e.g. In Table 1, Synthetic comes from GSM8k? And Human-annotated from MATH? And what about the Math-Shepherd? There should be a clear definition for each dataset with a unique name that then is used across the various experiments). Section 5 lacks clarity in general, but in particular it is very confusing what datasets are used for what.
>
> In Section 5.1 Dataset we introduce the base datasets to construct our synthetic data. GSM8k and MATH are two base datasets with math problems and corresponding reasoning chains. CG-H describes the data randomly sampled from the two datasets. FG-H introduces the synthetic datasets generated from questions in CG-H. Math-Shepherd includes automatically constructed process-wise supervision data using the Monte Carlo tree search method, where each step is assigned a binary label to indicate its correctness. We randomly sample a subset from it to test our models. To simplify the name, we still use its original name.
>
> For the hallucination detection task, we construct both synthetic and human-annotated datasets based on questions from the MATH dataset, and the CG-H (MATH) and FG-H (MATH) datasets are used to train and evaluate PRM and FG-PRM models, respectively. For the hallucination mitigation task, we employ the complete CG-H and FG-H datasets, along with Math-Shepherd, to train various reward models. Across all datasets, the training-to-testing split ratio is maintained at 95:5.
>
> We have updated our PDF, including all the details mentioned above in section 5.1 Dataset.
>
> > 6.  Right after Table 1 it is mentioned 700/100 samples from GSM8k and MATH for train/test. But then these numbers become 12,000, and it is not clear how (especially because they say the split of train/test is 95/5%).
>
> All the samples from GSM8k and MATH are utilized to generate synthetic data. For each dataset, we randomly sample 700 instances from the original training set and augment them to 12,000 instances by our automatic hallucination generation method, a prompt-based method whose details are in Section 4.2.1 Step 2. We add more details in the PDF.
>
> > 7.  Also in the OOD experiments, different datasets are used but they are not explained at all.
>
> In the OOD experiments, we have mentioned, “In these experiments, we train verifiers on the CG-H (GSM8K) and FG-H(GSM8K), and test them on the MATH dataset.” All concepts, including GSM8K, MATH, CG-H, and FG-H, have been introduced in Section 5.1, such as “GSM8K is a high-quality dataset consisting of grade school math problems designed to benchmark the reasoning abilities of language models. MATH is a large-scale dataset designed for probing and improving model reasoning, which includes human-written step-by-step solutions.”

---

> ### Author Response · Authors · 2024-12-24
> **Reply to Questions Cont.**
>
> > 8.  In Section 6, in the OOD paragraph “As detailed in Figure 4, the gap in CG-H (+0.3) is more significant than in FG-H (+0.1).” the numbers should be 0.03 and 0.01
>
> Thank you for pointing out the discrepancy. The numbers should be 0.03 and 0.01. We have revised this in Section 6 to ensure accuracy in the final version of the paper.
>
> > 9.  In Figure 3, if I correctly interpret y-axis to be accuracy (as you state in 5.3 by saying “Longformer and Llama3-8B achieve 94% and 58% accuracy on GSM8k and MATH”) then write “accuracy” as a label, and not “percentage”
>
> Thank you for bringing this to our attention. The y-axis should represent accuracy, and we have updated the label in Figure 3 from "percentage" to "accuracy" for clarity and consistency with Section 5.3.
>
> > 10.  In 4.2.2, “where the [sep] token represents the classification output at each reasoning step to indicate whether the following step y i contains the hallucination type t.” Instead of “following” it should be “previous”, I suppose
>
> Thank you for pointing this out. it should be "previous" rather than "following." We have revised the wording in Section 4.2.2 to accurately reflect this.
>
> > 11.  End of 5.1 you say “In inference processes, we sum all results as our final results”. This is not clear to me. Do you refer to Eq. 6? If yes then refer to it, or drop the sentence completely since it is already explained above.
>
> Thank you for your observation. The sentence refers to Equation 6. We have removed the sentence if deemed redundant since it is already explained earlier.
>
> > 12.  Spotted typos:
>
> Thank you for catching these typos. We have corrected them as follows:
> -   In Section 5.3, revise "verifiers become more evidence in the more complex MATH dataset" to "verifiers become more evident in the more complex MATH dataset."
> -   In Section 8, revise "we propose FG-PRM framework,introduces" to "we propose the FG-PRM framework, which introduces."
>
> We appreciate your attention to detail.
>
> > 13.  In the dataset you generate with specific hallucination types. What happens if more hallucinations are present at the same time (in a single step)? Is it possible or do you have a way of guaranteeing a single hallucination per step?
>
> Thank you for your question. We have manually verified the generated data by randomly sampling 100 examples from each category and confirmed that the data aligns with the intended categorization. Additionally, the presence of multiple hallucination types in a single step does not affect the training of our PRMs, as each PRM is designed to specifically judge whether a step contains the targeted hallucination type, independent of other potential types.
>
> > 14.  In the hallucination detection task, would a random classifier have 50% accuracy? If yes, how do the different methods compare to it? If not, what is the hardness of the task then? Could be good to explain this better and include the accuracy also, if you think it makes the discussion clearer.
>
> Thank you for raising this point. We focus on evaluating the performance of PRMs throughout the reasoning process. Thus, we evaluate the performance of PRMs on the instance level but not the step level. For each individual step, a random classifier would have 50% accuracy. However, given a reasoning process consisting of n steps, the accuracy of a random classifier drops significantly to (½)^n. Under this situation, it is obvious that the precision, recall, and F1 scores will be significantly lower than our results in Tables 1, 5, and 6.
>
> > 15.  About the datasets, if I correctly understand, you train the 6 models that compose a FG-PRM separately, so you have in practice 6 separate datasets, one per hallucination type?
>
> Thank you for your question. We train the six models that compose the FG-PRM separately, and each model is trained on its corresponding dataset, which contains examples specific to a particular hallucination type. This separation ensures that each model specializes in detecting one specific hallucination type effectively.
>
> > 16.  In section 4.2.1, Step 1, you say “To achieve this, we have developed a set of tailored rules for the Llama3-70B”. These rules are “prompts”, right? I would call them “prompts” then, and not “rules”, otherwise it sounds more formal than what it is.
>
> Thank you for your observation. The "rules" we refer to in Section 4.2.1 are indeed included in the prompts. We use the term "rules" because these instructions within the prompts guide Llama3-70B to generate synthetic data in a structured and controlled manner. However, we understand your concern about potential confusion and have rephrased it to "rules within the prompts" for greater clarity.

---

### Review · Reviewer_MezR · 2024-12-11

**Summary Of Contributions:**

This paper proposes to improve detection of "hallucinations" in LLMs by training a family of reward models, each of which is specialized to a particular type of "hallucination". In order to do so, the authors propose a new taxonomy of hallucinations, which they argue is more fine-grained than that used in prior work. Using this taxonomy, they then generate large sets of synthetic data by injecting a specific type of hallucination into a gold-standard reasoning chain with API-served, pretrained models. The authors then train their specialized reward models independently for each subset of the data; the final score of a candidate solution is then taken to be the sum of each reward model's score. Empirically, the authors find that the overall score forms a better signal for the correctness of the solution than those of course-grained reward models, although simply prompting pre-trained models appears to be competitive in some settings.

**Audience:**

Yes

**Broader Impact Concerns:**

None.

**Claims And Evidence:**

No

**Requested Changes:**

- Major: Training one reward model (which, in your setting, is basically a binary classifier) per hallucination type is one way to use fine-grained data. However, you could also imagine training a single, multi-class classifier. Prior work in the code generation literature (which shares a great deal of overlap with the type of reasoning tasks you consider here) has found that the score assigned by this single model to the "no error" label is a much stronger signal than when a binary model is trained on coarse-grained data [1]. I would like to see this included as a baseline in this paper, since it's otherwise unclear what the impact is of some particularities of your setup (e.g. that you don't normalize the reward).
- Major: Per equation 6 and the preceding discussion, your formal definition of FG-PRM makes it sound like it at time $i$ only has access to the input/prompt $x$ and the current "step" $y_i$; if this were the case you could of course not detect, for example, logical inconsistencies between $y_i$ and $y_{<i}$. There thus appears to be a mismatch between your verbal explanation of FG-PRM and the formal/precise definition given, which needs to be rectified.
- Major: Section 5.3 does not make it clear what data the PRM baseline is trained on in the non-Math Shepherd setting. (Section 5.1 appears to be stating only that you use FG-H for both testing and training in the *synthetic* evaluation section?) If it is trained on FG-H then this should be stated clearly, so that the reader does not think you (for example) simply lifted the number in the table from some previous work. (If it is trained on CG-H then these results are of course meaningless, as the FG-PRM is benefitting from having access to a much larger set of training data.)
- Minor: The discussion of the evaluation on "Human-annotated Data" is not very clear. You say that you use GPT-3.5 to generate solutions for 50 problems from MATH, and then later state that each hallucination type has 50 human-annotated reasoning chains. Assuming the types in your taxonomy are meant to be distinct, that would require 6 * 50 = 300 reasoning chains. Where do the extra 250 chains come from?
- Minor: Equation (6) does not normalize the reward w.r.t. the length of the answer. It seems to me that this means that long answers with low confidence may be assigned a higher reward than short sequences with high confidence, and vice versa. I think the paper would benefit greatly from an investigation into this phenomenon and whether or not it causes issues in your setup.
- Minor: The decision to train fine-grained vs. coarse-grained reward models appears to be orthogonal to whether the reward model is outcome-based or process-based. Have you considered training "FG-ORM" models? How do you expect such a model would do, compared to FG-PRM? This would be interesting to discuss in the paper.


[1] @misc{inala2022faultawareneuralcoderankers,
      title={Fault-Aware Neural Code Rankers},
      author={Jeevana Priya Inala and Chenglong Wang and Mei Yang and Andres Codas and Mark Encarnación and Shuvendu K Lahiri and Madanlal Musuvathi and Jianfeng Gao},
      year={2022},
      eprint={2206.03865},
      archivePrefix={arXiv},
      primaryClass={cs.PL},
      url={https://arxiv.org/abs/2206.03865},
}

**Strengths And Weaknesses:**

Strengths:
- The experiments are fairly extensive and, for the most part, presented well.
- The idea of training fine-grained reward models is presented very clearly (other than some major flaws with the formal definitions used; see requested changes)

Weaknesses:
- While the experiments are extensive, I do not think they target the right set of questions. In particular, I do not see the point of including experiments on synthetic data generated via the same method as your training data; of course the method you designed that synthetic data generation method for will benefit the most from it, and that says nothing about its performance in the real world.
- It is unclear how much of the reported gains in accuracy are due to the synthetic data generation process rather than the FG-PRM itself (see requested changes).
- The taxonomy appears completely arbitrary to me; some of your categories I would not consider to be "hallucinations" at all. The word "hallucination" is not synonyms with "mistake" or "error", and trying to make it so is the only reason why you need a taxonomy in the first place. This strikes me as an instance of "a solution in search of a problem".

---

> ### Author Response · Authors · 2024-12-24
> **Response to Questions**
>
> Thank you for your insightful suggestions and comments. We address your concerns and answer your questions below. All updates have been highlighted in red in the PDF.
>
> > 1.  While the experiments are extensive, I do not think they target the right set of questions. In particular, I do not see the point of including experiments on synthetic data generated via the same method as your training data; of course the method you designed that synthetic data generation method for will benefit the most from it, and that says nothing about its performance in the real world.
>
> Thank you for your feedback. In Table 1, we compare the performance of PRMs on the synthetic dataset with their performance on the human-annotated dataset. The human-annotated dataset consists of LLM-generated responses with hallucination types labeled by humans, making it a reasonable approximation of real-world cases. To avoid misinterpretation, we have refined the description of the data annotation process, as shown in the Human-annotated Data paragraph.
>
> Moreover, our results show that the performance of PRMs on the synthetic dataset closely aligns with their performance on the human-annotated dataset. This consistency demonstrates that the synthetic dataset serves as an effective proxy for real-world scenarios, validating the utility of our approach.
>
> > 2.  The taxonomy appears completely arbitrary to me; some of your categories I would not consider to be "hallucinations" at all. The word "hallucination" is not synonyms with "mistake" or "error", and trying to make it so is the only reason why you need a taxonomy in the first place. This strikes me as an instance of "a solution in search of a problem".
>
> Thank you for your thoughtful feedback. In NLP, "hallucination" is broadly defined as generated content that is nonsensical or unfaithful to the provided source [1]. While we understand your concerns, variations in the definition of hallucination exist across different natural language generation tasks. Our taxonomy is specifically designed for reasoning tasks and aims to provide a structured framework for analyzing distinct error patterns. The taxonomy is not intended to equate all errors with hallucinations but rather to enable a more granular analysis and inform targeted mitigation strategies. We believe this systematic approach is essential for addressing key challenges in model evaluation and training.
>
> To address your concern, we have revised the manuscript to better clarify the motivation for the taxonomy and the reasoning behind the terminology. We appreciate any additional suggestions you might have for improving this aspect of our work.
>
> [1] Ji, Ziwei, et al. "Survey of hallucination in natural language generation." ACM Computing Surveys 55.12 (2023): 1-38. https://dl.acm.org/doi/10.1145/3571730
>
> > 3.  Major: Training one reward model (which, in your setting, is basically a binary classifier) per hallucination type is one way to use fine-grained data. However, you could also imagine training a single, multi-class classifier. Prior work in the code generation literature (which shares a great deal of overlap with the type of reasoning tasks you consider here) has found that the score assigned by this single model to the "no error" label is a much stronger signal than when a binary model is trained on coarse-grained data [1]. I would like to see this included as a baseline in this paper, since it's otherwise unclear what the impact is of some particularities of your setup (e.g. that you don't normalize the reward).
>
> Thank you for your suggestion. We train a single, multi-classifier on a Longformer model. In the following table, we add the experiment as a baseline:
>
> | Detector         | Context Inconsistency | Logical Inconsistency | Instruction Inconsistency | Logical Error | Factuality Inconsistency | Fabrication | Average |
> |---|---|---|---|---|---|---|---|
> | FG-PRM (compact) | 0.402                 | 0.493                 | 0.481                     | 0.378         | 0.371                    | 0.574       | 0.450   |
> | FG-PRM           | 0.488                 | 0.549                 | 0.529                     | 0.398         | 0.422                    | 0.608       | 0.499   |
>
> | Verifier / Reward Model | GSM8k | MATH |
> |---|---|---|
> | Self-Consistency        | 0.88  | 0.48 |
> | FG-PRM                  | 0.94  | 0.57 |
> | FG-PRM (compact)        | 0.90  | 0.54 |
>
> We have updated the results in our PDF.  Our findings indicate a tendency for the model to predict the "no error" label. This bias is primarily due to the imbalance in the training data. Instances with a specific type of hallucination account for only one-sixth of the entire dataset. Additionally, in the multi-step reasoning process, only a few steps display specific hallucinations. As a result, models can achieve high accuracy by predominantly predicting "no error."
>
> About normalizing the reward, please refer to our reply to question 7.

---

> ### Author Response · Authors · 2024-12-24
> **Response to Questions Cont.**
>
> > 4.  Major: Per equation 6 and the preceding discussion, your formal definition of FG-PRM makes it sound like it at time i only has access to the input/prompt x and the current "step" yi; if this were the case you could of course not detect, for example, logical inconsistencies between yi and y<i. There thus appears to be a mismatch between your verbal explanation of FG-PRM and the formal/precise definition given, which needs to be rectified.
>
> Thank you for pointing this out. As shown in the input format described in Section 4.2.2, "y_1 [sep] y_2 [sep] ... y_L [sep]," each [sep] token represents the classification output at each reasoning step to indicate whether the previous step y_i​ contains the hallucination type t. This input structure ensures that the model has access to step i as well as all preceding steps, enabling it to detect logical inconsistencies between y_i​ and y_{<i}​. We have clarified this in the paper to avoid any misunderstanding.
>
> > 5.  Major: Section 5.3 does not make it clear what data the PRM baseline is trained on in the non-Math Shepherd setting. (Section 5.1 appears to be stating only that you use FG-H for both testing and training in the synthetic evaluation section?) If it is trained on FG-H then this should be stated clearly, so that the reader does not think you (for example) simply lifted the number in the table from some previous work. (If it is trained on CG-H then these results are of course meaningless, as the FG-PRM is benefitting from having access to a much larger set of training data.)
>
> Thank you for the suggestion. The PRM baseline is trained on CG-H. In other words, the performance on the GSM8K dataset is from the verifier trained on CG-H (GSM8K). It is the same for the performance on MATH. To show the effectiveness of our FG-PRM, we conduct a new experiment using the same FG-H to train another PRM. The experiment results are in the reply to question 3. The superior performance of FG-PRM demonstrates the effectiveness of our synthetic data generation and fine-grained method.
>
> > 6.  Minor: The discussion of the evaluation on "Human-annotated Data" is not very clear. You say that you use GPT-3.5 to generate solutions for 50 problems from MATH, and then later state that each hallucination type has 50 human-annotated reasoning chains. Assuming the types in your taxonomy are meant to be distinct, that would require 6 * 50 = 300 reasoning chains. Where do the extra 250 chains come from?
>
> Thank you for pointing this out. That was a typo in the manuscript. To clarify, for each hallucination type, we began by using ChatGPT (GPT-3.5-turbo-0125) to generate step-by-step solutions for 50 distinct problems from the MATH dataset. These solutions were then manually annotated, ensuring that each selected solution included at least one step exhibiting the target hallucination type. As a result, we obtained a total of 300 annotated reasoning chains, with 50 reasoning chains for each of the six hallucination types. We have updated the discussion to make this clearer in the paper.
>
> > 7.  Minor: Equation (6) does not normalize the reward w.r.t. the length of the answer. It seems to me that this means that long answers with low confidence may be assigned a higher reward than short sequences with high confidence, and vice versa. I think the paper would benefit greatly from an investigation into this phenomenon and whether or not it causes issues in your setup.
>
> Thank you for pointing this out. We realize that our explanation could have been clearer. To address your concern, we have revised the formula in Section 4.2.2 to clarify this aspect. Specifically, the reward scores represent the probability of correctness, and the summation in the equation is a log-sum, equivalent to the product of all probabilities.
>
> In this setup, if a step is correct, its probability of being correct is close to 1, which minimally affects the final result. As a result, the length of the answer does not influence the reward score. We hope this clarification addresses your concern.

---

> ### Author Response · Authors · 2024-12-24
> **Response to Questions Cont.**
>
> > 8.  It is unclear how much of the reported gains in accuracy are due to the synthetic data generation process rather than the FG-PRM itself (see requested changes).
> > Minor: The decision to train fine-grained vs. coarse-grained reward models appears to be orthogonal to whether the reward model is outcome-based or process-based. Have you considered training "FG-ORM" models? How do you expect such a model would do, compared to FG-PRM? This would be interesting to discuss in the paper.
>
> Thank you for the insightful comment and suggestion regarding FG-ORM models. While we appreciate the potential interest in exploring such experiments, we opted not to pursue them for the following reasons:
>
> 1.  Comparison Context: The comparison experiment between FG-ORM and FG-PRM would primarily reflect the difference between process-based and outcome-based reward models (PRM vs. ORM). This comparison would not directly address the core question of the effectiveness of the synthetic data generation process compared to FG-PRM itself.
>
> 2.  Process-Supervision Dependency: Our synthetic data generation method is inherently designed for process-supervision. To evaluate the impact of the synthetic data generation process separately, we would need to compare FG-PRM models trained on synthetic data versus those trained on a fully human-labeled dataset. However, generating human-labeled data for fine-grained process supervision is highly labor-intensive, as each reasoning step requires six hallucination labels.
>
> 3.  Effectiveness of Synthetic Data: The results in Table 2 highlight that FG-PRM models trained on synthetic data outperform PRM models trained on human-labeled data. The performance gains underscore the quality and utility of the synthetic data in enhancing model capabilities. This indirectly validates the effectiveness of our synthetic data generation method.

---

> > ### Comment · Reviewer_MezR · 2024-12-28
> >
> > Thank you your extensive response.
> >
> > I now understand the human data annotation process, and this does alleviate my concern about testing on synthetic data. I apologize for the misunderstanding; I just wanted to make sure that you were not prompting the model to make a certain type of mistake, which obviously would not then transfer to real data. That being said, even in your updated manuscript it is not clear to me at a first read that you do not prompt the model to make a certain type of mistake, but that you simply prompt it to solve the task and then discard the correct solutions and label the rest. For future revisions you might want to consider clarifying this further.
> >
> > Regarding training the baseline PRM on CG-H, my concern is that--unless I am mistaken--FG-H contains novel/synthetic variations of the chains in CG-H. Thus the presence of the fine-grained labels is not the only difference between your FG-PRM and the baseline; it has also been trained on much more data. It would be beneficial to compare against a baseline PRM which is trained with coarse-grained labels but the full set of chains in FG-H, including the synthetically generated variations. The experiment you've added, which I find very interesting and which I am grateful that you were able to add so quickly, does not quite capture this setting.

---

### Comment · Action_Editor_sti2 · 2024-12-18
**Discussion Phase**

Dear authors,

This is a kind reminder that we are currently in the middle of the discussion period. Please note that responses to reviewers must be submitted within one week from today.

Kind Regards, \
AE

---

### Decision · Action_Editor_sti2 · 2025-01-11

**Recommendation:** Reject

**Comment:**

The paper received a mixed consensus, with two out of three reviewers leaning towards rejection and one supporting acceptance. All reviewers praised the clarity of the taxonomical concepts and acknowledged the timeliness and relevance of the topic, noting its potential appeal to the TMLR audience. Additionally, reviewers appreciated the significant improvements made during the rebuttal period.

However, the manuscript still falls short of meeting the criteria for acceptance in its current form, particularly in terms of technical soundness. While the work addresses an important and compelling research question, additional evidence and analysis are needed to fully substantiate the claims and convince the community to adopt the proposed taxonomy and evaluation methodology.

To improve the submission for future consideration, the authors are encouraged to address the following key points:
- Provide detailed descriptions of hyperparameters, datasets, and experimental setups.
- Include statistical measures (e.g., standard deviations, error margins) to ensure the reproducibility and robustness of the results.
- Conduct experimental validation to support the choices made for the taxonomy and the annotation procedure.

**Audience:**

This work represents an initial effort to conceptualize a taxonomy of hallucination types in large language models (LLMs) and to develop a corresponding evaluation methodology for quantifying these types. The research question is both timely and relevant to the TMLR audience, with the potential to attract significant interest from subcommunities focused on generative models and language models.

**Claims And Evidence:**

Reviewers raised important concerns about the technical soundness of the work. In particular, two key contributions of the paper, the taxonomy of hallucination types and the corresponding dataset annotation procedure, are inadequately supported by experiments or a thorough discussion of their underlying motivations. These issues remained unaddressed during the rebuttal period. Regarding the taxonomy, while it appears reasonable, its validity as a sound choice is unclear, as no methodological validation or supporting analysis has been provided. For the annotation methodology, there are notable shortcomings, particularly in the selection of annotators (a small pool of three students), limited diversity (three graduate students), and lack of statistical consistency (only 79% agreement). This undermines the quality and soundness of the overall annotation process.

Beyond concerns about the accuracy of the main claims, two of the three reviewers also expressed doubts about the soundness and reproducibility of the experimental methodology. Specific issues include the absence of statistical analysis of the results and insufficient details about hyperparameter configurations.

**Resubmission Of Major Revision:**

The authors may consider submitting a major revision at a later time.